# A Comprehensive Survey on Inverse Constrained Reinforcement Learning: Definitions, Progress and Challenges

**Guiliang Liu**[1]                                                               *liuguiliang@cuhk.edu.cn*

**Sheng Xu**[1]                                                               *shengxu1@link.cuhk.edu.cn*

**Shicheng Liu**[3]                                                               *sfl5539@psu.edu*

**Ashish Gaurav**[2,4]                                                               *ashish.gaurav@uwaterloo.ca*

**Sriram Ganapathi Subramanian**[2,4]                                                               *s2ganapa@uwaterloo.ca*

**Pascal Poupart**[2,4]                                                               *ppoupart@uwaterloo.ca*

[1] *School of Data Science, The Chinese University of Hong Kong, Shenzhen,*
[2] *Cheriton School of Computer Science, University of Waterloo,*
[3] *Pennsylvania State University,*
[4] *Vector Institute*

**Reviewed on OpenReview:** *https://openreview.net/forum?id=WUQsBiJqyP*

## Abstract

Inverse Constrained Reinforcement Learning (ICRL) is the task of inferring the implicit constraints that expert agents adhere to, based on their demonstration data. As an emerging research topic, ICRL has received considerable attention in recent years. This article presents a categorical survey of the latest advances in ICRL. It serves as a comprehensive reference for machine learning researchers and practitioners, as well as starters seeking to comprehend the definitions, advancements, and important challenges in ICRL. We begin by formally defining the problem and outlining the algorithmic framework that facilitates constraint inference across various scenarios. These include deterministic or stochastic environments, environments with limited demonstrations, and multiple agents. For each context, we illustrate the critical challenges and introduce a series of fundamental methods to tackle these issues. This survey encompasses discrete, virtual, and realistic environments for evaluating ICRL agents. We also delve into the most pertinent applications of ICRL, such as autonomous driving, robot control, and sports analytics. To stimulate continuing research, we conclude the survey with a discussion of key unresolved questions in ICRL that can effectively foster a bridge between theoretical understanding and practical industrial applications. The papers referenced in this survey can be found at https://github.com/Jasonxu1225/Awesome-Constraint-Inference-in-RL.

## 1 Introduction

To ensure the reliability of a Reinforcement Learning (RL) algorithm within safety-critical applications, it is crucial for the agent to have knowledge of the underlying constraints. However, in many real-world tasks, the constraints are often unknown and difficult to specify mathematically, particularly when these constraints are time-varying, context-dependent, and inherent to the expert's own experience. For example, Figure 1 shows a contemporary example of a highway merging task, where the ideal constraints depend on the traffic or road conditions as well as the weather.

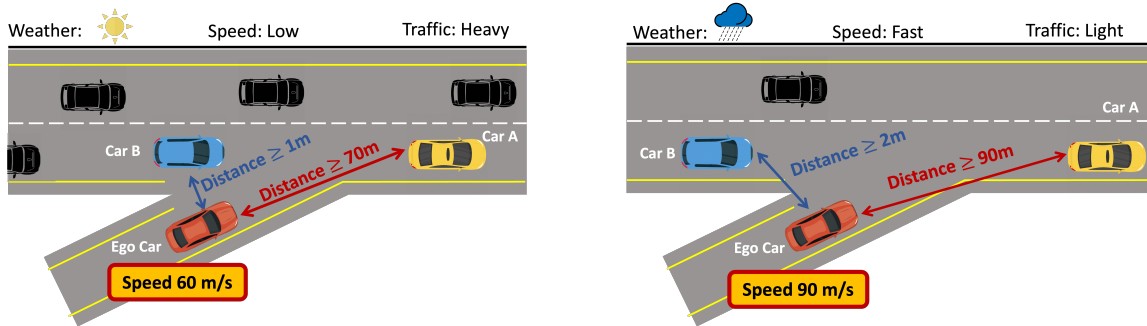

Figure 1: An example of the context-sensitive car distance constraint between vehicles during a merge on the highway. Under proper weather conditions, when vehicle speed is relatively low and traffic congestion is high, the distance between cars can be reduced. However, in adverse weather conditions, when vehicles are moving fast and traffic is sparse, it becomes necessary to increase the distance between cars to ensure safety.

An effective approach to resolve the above challenges is Inverse Constrained Reinforcement Learning (ICRL), which infers the implicit constraints obeyed by expert agents, utilizing experience collected from both the environment and the observed demonstration dataset. These constraints, learned through a data-driven approach, can effectively generalize across multiple environments, thereby providing a more comprehensive explanation of the expert agents' behavior and facilitating safety control in downstream applications.

To infer the underlying constraints, ICRL alternates between updating an imitating policy with Constrained Reinforcement Learning (CRL) and learning a constraint function via Inverse Constraint Inference (ICI), until the imitation policy can reproduce expert demonstrations. Figure 2 shows an illustrative example of the learning process of ICRL in a grid-world environment where the agent seeks the shortest path from the start state to the goal state. In the absence of any constraint, the agent optimizes a policy that initially returns a direct path from the start state to the goal state (Figure 2 Round 1). However, since this path does not imitate the expert path, the agent infers that the orange region must be infeasible, given that it is not visited by the expert demonstration. In Round 2, the agent optimizes a revised policy subject to this constraint that produces a new path. Once again, the agent infers that the orange region must be infeasible since it is not visited by the expert demonstration. The process keeps on alternating between constrained policy optimization and constraint inference until the resulting policy imitates expert demonstrations.

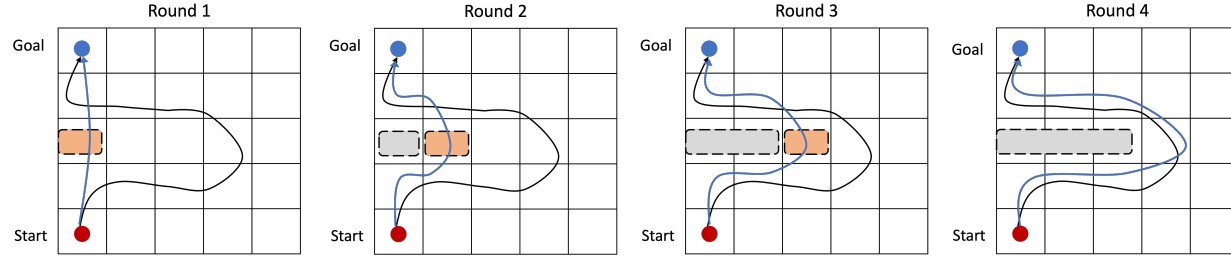

Figure 2: A running example of ICRL, which alternates between policy updates and constraint inference in each round. The expert policy and the imitation policy are represented by the black and blue curves, respectively. The newly inferred constrained region in each round is highlighted in orange, while the constrained region inferred in previous rounds is depicted in gray.

ICRL, as an evolving research area, has received significant attention in recent years. This article offers a comprehensive introduction to ICRL including recent advancements.

## 1.1 The Significance of this Survey

The survey makes the following contributions.

Table 1: A structured summary of the key ICRL algorithms that are reviewed in this article.

| Reference | Method& Framework | State/Action- Space | Environment Dynamics | Dataset- Coverage | Agent- Numbers |
|---|---|---|---|---|---|
| (Scobee & Sastry, 2020) | Maximum Entropy | Discrete | Deterministic | Full | Single |
| (Malik et al., 2021) | Maximum Entropy | Continuous | Deterministic | Full | Single |
| (McPherson et al., 2021) | Maximum Causal Entropy | Discrete | Stochastic | Full | Single |
| (Baert et al., 2023) | Maximum Causal Entropy | Continuous | Stochastic | Full | Single |
| (Gaurav et al., 2023) | Deep Constraint Correction | Continuous | Stochastic | Full | Single |
| (Xu & Liu, 2024b) | Robust Optimization | Continuous | Stochastic | Full | Single |
| (Papadimitriou et al., 2023) | Bayesian Framework | Discrete | Stochastic | Partial | Single |
| (Liu et al., 2023) | Variational Inference | Continuous | Stochastic | Partial | Single |
| (Subramanian et al., 2024) | Confidence-Aware Estimation | Continuous | Stochastic | Partial | Single |
| (Xu & Liu, 2024a) | Generative Model | Continuous | Stochastic | Partial | Single |
| (Quan et al., 2024) | Distribution Correction Estimation | Continuous | Stochastic | Partial | Single |
| (Papadimitriou & Brown, 2024) | Preference-Based Estimation | Continuous | Stochastic | Partial | Single |
| (Park et al., 2019) | Bayesian Framework | Discrete | Stochastic | Full | Single |
| (Jang et al., 2023) | Reward Decomposition | Continuous | Stochastic | Full | Single |
| (Lindner et al., 2024) | Linear Programming | Continuous | Deterministic | Full | Multiple |
| (Kim et al., 2023) | Multi-Task IRL | Continuous | Deterministic | Full | Multiple |
| (Qiao et al., 2023) | Multi-Modality RL | Continuous | Deterministic | Full | Multiple |
| (Liu & Zhu, 2022; 2023a) | Maximum Likelihood | Continuous | Stochastic | Full | Multiple |

**In-Depth Guide to ICRL.** More specifically, we introduce the forward and backward procedures of ICRL within the framework of a Constrained Markov Decision Process (CMDP). To substantiate the rationale behind ICRL, we highlight fundamental differences in comparison to Inverse Reinforcement Learning (IRL) and Inverse Optimal Control (IOC).

**Overview of Recent Advancements.** To illustrate the recent developments in a structured format, Table 1 presents the existing ICRL methodologies under various learning contexts and environments. Specifically: 1) In **deterministic** environments, we present the Maximum Entropy (MEnt) ICRL approach, which investigates both discrete and continuous domains. 2) In **stochastic** environments, our focus shifts to model the Maximum Causal Entropy (MCEnt) and soft constraints. 3) To manage demonstration datasets that can only encapsulate **partial knowledge** of the environment, we provide an overview of the distribution-based, data-augmented, and offline ICRL techniques. 4) We introduce approaches to learn constraints and rewards **simultaneously**. 5) This survey explores ICRL in the context of **multi-agent settings**, detailing how to infer shared or individual constraints based on different types of agents involved.

**Motivation for Future Research.** To pave the way for future research, our survey encompasses evaluation benchmarks under discrete, virtual, and realistic environments. Additionally, we discuss the practical usage of ICRL by introducing its potential applications in diverse fields such as autonomous driving, robot control, and sports analytics. Finally, we outline some open problems and challenges of ICRL, hereby highlighting the areas where future research can contribute significantly and bridge existing gaps.

## 1.2 Organization of Contents

As ICRL is an emerging field, this article is intended to serve as a comprehensive guide for readers keen to learn about it. The structure of this article is outlined as follows: 1) In Section 2, we provide a thorough introduction to the *foundations and terminologies* of ICRL. This includes the mathematical definitions of RL, Constrained RL, and ICRL, as well as the methods employed to regularize the learned constraints. 2) Section 3 delves into the *research topics related to ICRL*, such as Inverse Reinforcement Learning (IRL) and Inverse Optimal Control (IOC). We highlight their fundamental differences from ICRL, thereby underscoring the unique value of ICRL. 3) Section 4 and Section 5 explore recent advancements in ICRL. This includes discussions about constrained inference algorithms derived from *deterministic and stochastic environments*. Section 6 and Section 8 describe recent progress in ICRL from *partial demonstration data* and *multiple agents* in the environment. 4) Section 7 investigates approaches to *simultaneously infer constraints and rewards*. 5) Section 9 presents the *benchmarks and real-world applications* of ICRL. 6) Section 10 concludes this survey and outlines important *challenges and open questions* for future ICRL research.

## 2 Background and Notation

In this section, we establish our notation and provide the essential background for a better understanding of the remainder of the survey. A synopsis of the primary notation and abbreviations is available in Table 2.

Table 2: The main notation and abbreviations throughout the paper.

| Symbols and abbreviations | Meaning |
|---|---|
| $\mathcal{M}, \mathcal{M}_c$ | Markov Decision Process (MDP) and Constrained MDP. |
| $s, a, r$ | State, action, and reward in an MDP. |
| $c(s,a), \epsilon$ | Cost, and the corresponding threshold in a constraint. |
| $\gamma, \mu_0$ | Discount factor and the initial state distribution (i.e., $s_0 \sim \mu_0$). |
| $\mathcal{D}_E$ | Expert demonstrations dataset. |
| $\pi_E, \hat{\pi}, \Pi$ | Expert policy, nominal (i.e., imitation) policy, and the set of policies. |
| $\tau_E, \hat{\tau}, \Xi$ | Expert trajectory, nominal trajectory, and the set of trajectories. |
| $\rho^\pi$ | Occupancy measure in accordance with policy $\pi$. |
| $\phi(s,a)$ | The feasibility of performing actions under a given state. |
| $k$ | Labels for the agent identity. |
| $\omega, \theta, \psi$ | Model parameters for the constraint, policy and density models. |
| $Q(s,a), V(s)$ | State-action and state-only value functions. |
| $\mathcal{H}(\pi(\tau)), \mathcal{H}(\boldsymbol{a}_{0:T}\|\boldsymbol{s}_{0:T})$ | Trajectory entropy and causal entropy. |
| MEnt | Maximum Entropy. |
| MCEnt | Maximum Causal Entropy. |
| CRL | Constrained Reinforcement Learning. |
| ICI | Inverse Constraint Inference. |
| IRL | Inverse Reinforcement Learning. |
| ICRL | Inverse Constrained Reinforcement Learning. |

Specifically, we use lower-case letters (e.g., $s$) to denote scalars, bold lower-case letters to denote vectors (e.g., $\boldsymbol{s}$) and bold upper-case letters (e.g., $\boldsymbol{S}$) to denote matrices. In the rest of this section, we introduce key definitions for the task of Reinforcement Learning (Section 2.1), Constrained Reinforcement Learning (Section 2.2), and Inverse Constrained Reinforcement Learning (Section 2.3) as well as constraint regularization (Section 2.4) methods.

### 2.1 Reinforcement Learning

Reinforcement learning (RL) algorithms are generally based on an episodic Markov Decision Process (MDP) $\mathcal{M}$ (Sutton & Barto, 2018), which can be defined by a tuple $(\mathcal{S}, \mathcal{A}, p_{\mathcal{R}}, p_{\mathcal{T}}, \gamma, T, \mu_0)$ where: 1) $\mathcal{S}$ and $\mathcal{A}$ denote the space of states and actions. 2) $p_{\mathcal{T}}(s'|s,a)$ and $p_{\mathcal{R}}(r|s,a)$ define the transition and reward distributions. 3) $\gamma \in [0,1)$ is the discount factor. 4) $T \in [0, \infty)$ defines the planning horizon. 5) $\mu_0 = p(s)$ denotes the initial state distribution. In principle, this MDP terminates at a time step $T$, though this planning horizon $T$ is not fixed. For example, a game may 1) terminate when the agent reaches a terminating or goal state, or 2) assign a terminating probability associated with each state.

In an episodic MDP $\mathcal{M}$, the objective is to solve the sequential decision problem to maximize the (discounted) cumulative reward by learning a policy $\pi(a|s)$. Given that the literature on ICRL commonly adopts the Maximum Entropy (MEnt) framework, our survey aligns with the ICRL setting by focusing on the MEnt RL objective (Haarnoja et al., 2017). The MEnt RL objective can be described as follows::

$$J_\pi = \arg\max_\pi \mathbb{E}_{p_{\mathcal{R}}, p_{\mathcal{T}}, \pi, \mu_0} \left[ \sum_{t=0}^{T} \gamma^t r_t(s_t, a_t) \right] + \frac{1}{\alpha} \mathcal{H}(\pi) \tag{1}$$

where $\mathcal{H}(\pi)$ represents the policy entropy weighted by $\frac{1}{\alpha}$. Incorporating such an entropy regularizer offers several key advantages: 1) It can appropriately model the bounded rationality in human behaviors

(see Section 2.4). 2) It leads to a soft representation of the optimal policy, which can better model the sub-optimal behaviors and is more robust to stochasticity in the environment. 3) Depending on the problem settings, it can represent either trajectory-level entropy $\mathcal{H}(\pi(\tau)) = -\mathbb{E}_{\pi(\tau)}[\log \pi(\tau)]$ (see Sections 4.1 and 4.2) or causal entropy (i.e., discounted cumulative step-wise entropy (Bloem & Bambos, 2014)) $\mathcal{H}(\boldsymbol{a}_{0:T}|\boldsymbol{s}_{0:T}) = -\mathbb{E}_{\pi,p_{\mathcal{T}},\mu_0}[\sum_{t=0}^{T} \gamma^t \log \pi(a_t|s_t)]$ (see Section 5.1 and 5.2), accommodating the dynamics in both deterministic and stochastic environments. In the decision process, the agent generates a trajectory $\tau^\pi = [s_0, a_0, ..., a_{T-1}, s_T]$ and $p(\tau^\pi) = p(s_0) \prod_{t=0}^{T-1} \pi(a_t|s_t) p_{\mathcal{T}}(s_{t+1}|s_t, a_t)$. In this paper, we use $\Xi_{\mathcal{M}}$ to denote the set of trajectories in the MDP $\mathcal{M}$.

## 2.2 Constrained Reinforcement Learning

Constrained Reinforcement Learning (CRL) typically considers a constrained optimization problem under a Constrained Markov Decision Processes (CMDPs) $\mathcal{M}_c = (\mathcal{S}, \mathcal{A}, p_{\mathcal{R}}, p_{\mathcal{T}}, \{(p_{\mathcal{C}_i}, \epsilon_i)\}_{\forall i}, \gamma, T, \mu_0)$ which augments the original MDP $\mathcal{M}$ by adding constraints. $p_{\mathcal{C}_i}(c|s, a)$ denotes a stochastic constraint function [1] with an associated bound $\epsilon_i$, where $i$ indicates the index of a constraint, and the costs are positive and bounded, i.e., $c(s, a) \in [0, C_{\max})$. In this context, CRL agents consider a constrained optimization problem.

**Discounted Cumulative Constraints.** We consider a CRL problem where the goal is to find a policy $\pi$ that maximizes expected discounted rewards under a set of cumulative soft constraints:

$$\arg\max_{\pi} \mathbb{E}_{p_{\mathcal{T}},\pi,\mu_0} \Big[ \sum_{t=0}^{T} \gamma^t r(s_t, a_t) \Big] + \frac{1}{\alpha}\mathcal{H}(\pi) \tag{2}$$

$$\text{s.t.} \quad \mathbb{E}_{p_{\mathcal{T}},\pi,\mu_0} \Big[ \sum_{t=0}^{T} \gamma^t c_i(s_t, a_t) \Big] \leq \epsilon_i \ \forall i \in [1, I]$$

where $r(s_t, a_t) = \mathbb{E}_{p_{\mathcal{R}}}(r_t|s_t, a_t)$ and $c_i(s_t, a_t) = \mathbb{E}_{p_{\mathcal{C}_i}}(c_t|s_t, a_t)$. This formulation is most useful under an infinite decision horizon ($T = \infty$), where the constraints consist of bounds on the expectation of discounted cumulative cost values. Assuming $\epsilon_i > 0$, formula (2) can effectively represent *soft* constraints since it permits visiting some state-action pairs with high cost $c_i(\tilde{s}, \tilde{a}) \gg \epsilon_i$ as long as the chance of getting there is small, and the expectation of the discounted additive cost is smaller than the threshold ($\epsilon_i$).

Assuming the Markov chain induced by transition $p_{\mathcal{T}}(s_{t+1}|s_t, a_t)$ and policies $\pi \in \Pi$ to be irreducible and aperiodic (thus, the visitation frequency follows a unique stationary distribution), the constraint in Equation (2) can be equivalently represented as:

$$\mathbb{E}_{\rho_{\mathcal{M}}^\pi(s,a)} \Big[ c_i(s, a) \Big] \leq \epsilon_i \ \forall i \in [1, I] \tag{3}$$

with the normalized occupancy measure:

$$\rho_{\mathcal{M}}^\pi(s, a) = (1 - \gamma) \sum_{t=0}^{T} \gamma^t P_{\mathcal{M}}^\pi(s_t = s, a_t = a) \tag{4}$$

Here $P_{\mathcal{M}}^\pi(s_t = s, a_t = a)$ defines the probability of reaching $s$ at time step $t$ by implementing policy $\pi$ in the MDP $\mathcal{M}$. This constraint representation shows the constraint is linear (and thus convex) with respect to $\rho_{\mathcal{M}}^\pi$ (instead of $\pi$). Based on the occupancy measure, the CRL problem in 2 can be represented as:

$$\arg\max_{\pi} \mathbb{E}_{\rho_{\mathcal{M}}^\pi(s,a)} \Big[ r(s, a) - \frac{1}{\alpha} \log \pi(a|s) \Big] \tag{5}$$

$$\text{s.t.} \quad \mathbb{E}_{\rho_{\mathcal{M}}^\pi(s,a)} \Big[ c_i(s, a) \Big] \leq \epsilon_i \ \forall i \in [1, I]$$

where we slightly modify the original objective 2 by transforming the entropy into causal entropy. Although this optimization problem is non-convex, its Lagrange dual has no duality gap (Paternain et al., 2019).

---

[1] In the context of a deterministic environment, $p_{\mathcal{C}_i}(c|s, a)$ can be simplified as $c_i(s, a)$ which uniquely determines the cost based on a state-action pair.

Hence, we can transform the CRL problem 5 into its dual form:

$$\arg\min_{\pi} -\mathbb{E}_{\rho_{\mathcal{M}}^{\pi}(s,a)}\left[r(s,a) - \frac{1}{\alpha}\log\pi(a|s)\right] + \sum_{i}\lambda_i^*\left[\mathbb{E}_{\rho_{\mathcal{M}}^{\pi}(s,a)}\left(c_i(s,a)\right) - \epsilon_i\right] \tag{6}$$

where $\lambda_i^*$ denotes the optimal Lagrange multiplier for the $i^{th}$ constraint. For simplicity, let's consider only one constraint, and we have:

$$\arg\min_{\pi} -\mathbb{E}_{\rho_{\mathcal{M}}^{\pi}(s,a)}\left[r(s,a) - \lambda^*c(s,a) - \frac{1}{\alpha}\log\pi(a|s)\right] - \lambda^*\epsilon \tag{7}$$

**Trajectory-based Constraints.** An alternative approach is to define constraints on trajectories (i.e., sequences of state-action pairs) denoted by $\tau$:

$$\arg\max_{\pi} \mathbb{E}_{p_{\mathcal{R}},p_{\mathcal{T}},\pi}\left[\sum_{t=0}^{T}\gamma^t r_t\right] + \frac{1}{\alpha}\mathcal{H}(\pi) \tag{8}$$

$$\text{s.t.} \quad \mathbb{E}_{\tau\sim(p_{\mathcal{T}},\pi),p_{\mathcal{C}_i}}\left[c_i(\tau)\right] \leq \epsilon_i \ \forall i \in [1, I]$$

Depending on how we define the trajectory cost $c(\tau)$, the trajectory constraint can be more expressive than the cumulative constraint, for example, Transformers (Vaswani et al., 2017) can be applied to predict the trajectory cost in a way that does not decompose into a sum of step-wise costs. This trajectory constraint is useful in the episodic MDP with a limited (but not necessarily fixed) planning horizon.

**Hard versus Soft Constraints.** When $\epsilon$ is set to the minimum cost among all trajectories (i.e., $c_{min} = \arg\min_{\tau} c_i(\tau)$, where previous work commonly sets $c_{min} = 0$ ), it imposes a hard constraint on the problem. This ensures that the agent incurs a minimal cost for all trajectories. Conversely, when $\epsilon$ is chosen to be greater than $c_{min}$, it signifies a soft constraint. This scenario permits the agent to incur higher costs than $\epsilon$ for some trajectories, provided that other trajectories incur costs less than $\epsilon$ and the expected value of the costs does not exceed $\epsilon$.

### 2.3 Inverse Constrained Reinforcement Learning

In practice, instead of observing the constraint costs, we often have access to expert demonstrations $\mathcal{D}_E = \{\tau_E\}$ generated by the expert (with policy $\pi_E$) that satisfy the underlying constraints. The class of methods that aim to learn a policy from expert demonstrations is named apprenticeship learning (Abbeel & Ng, 2004). Among these methods, behavior cloning follows a supervised learning approach where the learner directly maps states to actions by observing the expert's demonstrations. However, this approach is sensitive to prediction errors. Divergence from the expert trajectory could lead to a cascade of errors in a sequential decision-making prob-

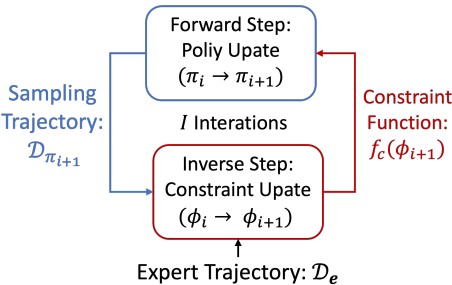

Figure 3: The flowchart of ICRL.

lem. Furthermore, it is particularly challenging to transfer the imitation policy to a new environment with different dynamics from the learning environment. Beyond learning from expert demonstrations, a growing body of research emphasizes learning from human interventions (Li et al., 2022; Basich et al., 2023; Spencer et al., 2022). This approach addresses the limitations of demonstration-based learning, which is restricted to scenarios where humans can directly operate the system to produce demonstrations.

Striving for robustness and generalizability, in this study, we focus on the preference modeling approach where the agent must first recover the rewards optimized and constraints respected by expert agents, and imitate experts by optimizing the CRL objective under these constraints. This is a challenging task since there might be various equivalent combinations of reward distributions and constraints that can explain the same expert demonstrations (Ziebart et al., 2008). Striving for identifiability, ICRL algorithms simplify the problem by assuming that rewards are observable, and the goal is to *recover only the constraints that best explain the*

*expert data* (Scobee & Sastry, 2020). To better elucidate the objectives of ICRL, we discuss the technical differences between ICRL and other relevant methods, including Inverse Reinforcement Learning (IRL), Inverse Optimal Control (IOC), and algorithms for simultaneous recovery of both rewards and constraints in Section 3. The inference process of ICRL often involves alternating between updating an imitating policy and updating a constraint function. Figure 3 summarizes the main procedure of ICRL. ICRL solvers involve learning the cost functions $c$ from an expert demonstration dataset $\mathcal{D}_E$. This is essentially solving a tri-level optimization problem (Kim et al., 2023):

$$\max_c \max_\lambda \min_\pi \mathbb{E}_{(s_E,a_E)\sim\mathcal{D}_E}\left[r(s_E,a_E) - \lambda c(s_E,a_E) - \frac{1}{\alpha}\log\pi(a_E|s_E)\right] -$$
$$\mathbb{E}_{(s,a)\sim\rho^\pi_\mathcal{M}(s,a)}\left[r(s,a) - \lambda c(s,a) - \frac{1}{\alpha}\log\pi(a|s)\right] - \lambda\epsilon \tag{9}$$

where the cost function $c$, the Lagrange parameter $\lambda$ and the policy function $\pi$ are subject to optimization at each level.

## 2.4 Regularizing the Learned Constraints

ICRL is essentially an ill-posed problem since there exist different constraints that can equivalently explain the expert demonstration, and thus the true constraints are not uniquely identifiable, as illustrated in Figure 4. This characteristic makes identifying the true underlying constraints a difficult task. To mitigate these challenges and enhance the identification of the optimal constraints, an effective method is to add regularization into constraint learning. Popular constraint regularization methods include:

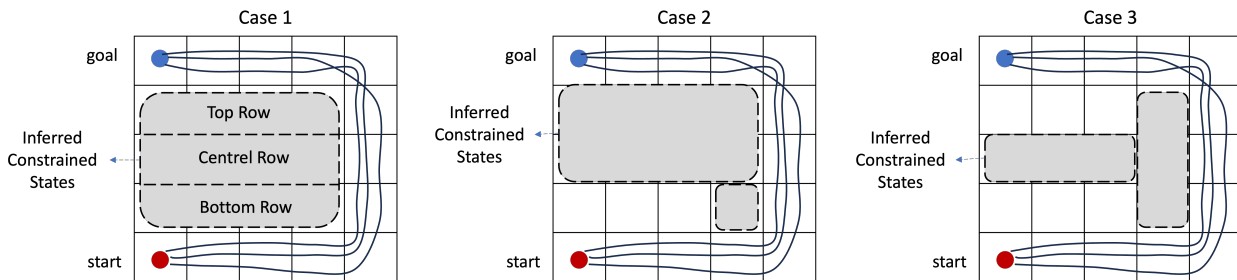

Figure 4: Examples of the ICRL solutions. In these illustrations, the initial location and the final destination are represented by red and blue circles, respectively. The expert demonstrations, signified by dark curves, are directly observable. The three distinct constraints recovered by ICRL algorithms, highlighted as gray regions, provide valid explanations for expert behaviors.

**Minimal Constraints.** An effective regularization approach involves limiting the complexity of the constraint. In a discrete domain, this can be understood as identifying the constraint set (i.e., the unsafe set) $\mathcal{C}$, which includes only the minimum number of state-action pairs that are critical for explaining the expert behaviors (Scobee & Sastry, 2020). For example, in Figure 4, the central row constraint is most critical since it explains why the agent intentionally avoids certain regions by taking a greater number of steps, consequently diminishing the cumulative rewards. In contrast, the constraints depicted in the top and bottom rows are less significant as they do not exert a substantial influence on the agents' behavior. In the continuous domain, constraining the complexity often implies enhancing the sparsity of the cost in the underlying CMDP (Malik et al., 2021). Standard sparsity regularizers (e.g., the $\ell_1$ norm) can be directly incorporated into the objective of constraint inference.

**Bounded Rationality.** In decision-making theory, the concept of bounded rationality (Simon, 1990) suggests that individuals tend to choose satisfactory decisions rather than optimal ones. Regarding the task of ICRL, we assume that expert agents employ a mixed strategy (i.e., $\pi_E(a|s) \geq 0, \forall(s,a) \notin \mathcal{C}$) where the probability of selecting sub-optimal actions is always greater than zero unless the state-action pair $(s,a)$ is infeasible. A useful representation of this mixed strategy can be attained by optimizing the policy entropy, which can be achieved by maximizing either the trajectory-level entropy in deterministic environments (see

Section 4) or the causal entropy in stochastic environments (see Section 5). Bounded rationality is referred to as indirect regularization because the entropy primarily affects the policy representation, which in turn influences constraint estimation.

**Interpretable Constraints.** In the pursuit of Explainable AI (XAI), it is essential for the constraints to be interpretable, such as identifying which feature most significantly impacts the cost function. Besides, XAI can necessitate that the cost function possesses physical significance. For instance, Liu et al. (2023) defined the cost function as $c(s, a) = 1 - \phi(s, a)$ and Qiao et al. (2023) defined the cost function as $c(s, a) = -\log \phi(s, a)$, where the feasibility function $\phi(s, a)$ represents the probability that executing action $a$ in state $s$ is safe. Besides, XAI techniques can be incorporated into the training or design of the cost function to understand the reasons behind the safety of a particular action or state.

## 3 Related Topics

In this section, we cover the research topics that are relevant yet distinct from ICRL. While studying these topics is not our primary focus, we provide an overview of their key definitions and algorithms, and most importantly, highlight the crucial differences with ICRL in order to aid readers in gaining a more comprehensive understanding of our study.

### 3.1 Inverse Reinforcement Learning

Inverse Reinforcement Learning (IRL) is the method for deducing the reward function within an MDP based on observed expert demonstrations. Multiple algorithmic frameworks have been developed for addressing the IRL problem. These include methods like maximum marginal (Ng & Russell, 2000), Maximum Entropy (MEnt) (Ziebart et al., 2008), Bayesian inference (Ramachandran & Amir, 2007), and adversarial learning techniques (Fu et al., 2017).

To provide a concrete example, MEnt IRL, which is among the most extensively studied IRL algorithms, formulates the objective function as a two-player max-min game (Garg et al., 2021):

$$\max_{r \in \mathcal{R}} \min_{\pi \in \Pi} L(\pi, r) = \mathbb{E}_{\rho_E}[r(s, a)] - \mathbb{E}_{\rho_\pi}[r(s, a)] - \mathcal{H}(\pi) - \psi(r) \tag{10}$$

where $\rho_E$ and $\rho_\pi$ refer to occupancy measures derived by the expert and imitation policies, $\mathcal{H}$ denotes the entropy and $\psi$ is a convex reward regularizer. Given that this is a concave-convex max-min objective, the Lagrange duality gap between the dual problem and the original one is effectively closed. This structure allows the use of established optimization techniques (e.g., Karush-Kuhn-Tucker (KKT) conditions) for deriving the optimal policy representation:

$$\pi(a_t|s_t) = \frac{\exp[Q^{soft}(s_t, a_t)]}{\int \exp[Q^{soft}(s_t, a)]\mathrm{d}a} \tag{11}$$

where the function $Q^{soft}$ satisfies the soft Bellman equation:

$$Q^{soft}(s_t, a_t) = r(s_t, a_t) + \gamma \mathbb{E}_{s'_{t+1} \sim p(\cdot|s_t, a_t)} \Big[ \log \sum_{a_{t+1}} \exp Q^{soft}(s_{t+1}, a_{t+1}) \Big] \tag{12}$$

Under this formulation, the goal is to find the reward function that can maximize the likelihood of generating expert demonstrations, i.e., the corresponding objective is

$$\arg\max_{r} \mathbb{E}_{\tau_E \in \mathcal{D}_E} \log \Big[ \prod_{t \in [0,T]} \pi(a_{E,t}|s_{E,t}) \Big] \tag{13}$$

A notable line of research, constrained inverse reinforcement learning (CIRL), integrates predefined constraints into the IRL framework. CIRL enables learning behaviors from expert demonstrations while ensuring adherence to these constraints (Schlaginhaufen & Kamgarpour, 2023; Ding & Xue, 2022; Renard, 2023). However, this line of research does not address the challenge of inferring the constraints.

**Comparing IRL versus ICRL:** Based on the tri-level optimization objective for ICRL (Eq. 9), a recent study (Hugessen et al., 2024) explored whether we can 1) apply an IRL algorithm to recover $\bar{r}(s,a) = r(s,a) - \lambda^* c(s,a)$ and 2) learn an imitation policy by directly maximizing the cumulative rewards $\mathbb{E}[\sum_{t=0}^{T} \gamma^t \bar{r}(s_t, a_t)]$ without considering the constrained optimization objective. While this simplification stabilizes the learning process, the reward learning method poses challenges in generalizing the learned constraints.

**Generalization Differences.** Within the frameworks of IRL and ICRL, the inferred reward and constraint functions are expected to generalize across environments. In other words, once we have learned reward or constraint functions, the goal is to reuse them in other environments. We show with a simple example that the generalization achieved by reward functions and constraint functions is different in new environments.

Consider a simple shortest path problem (Figure 5) where the shortest path goes through state $s^c$ and the second shortest path avoids state $s^c$. Suppose that the reward function is known and the shortest path earns a reward of 10 while the second shortest path earns a reward of 9. Suppose that the expert demonstrations always follow the second shortest path. Hence, we can learn a constraint that makes $s^c$ infeasible or we can learn an additional reward term that gives a penalty of $-1 - \eta$ (where $\eta > 0$). In both cases, the learned constraint and reward functions induce optimal policies that are consistent with the expert behavior.

Let's consider a new environment with the same shortest and second shortest paths, but a different reward function. The shortest path still earns a reward of 10, but the second shortest path earns a reward of $9 - 2\eta$. If we apply the learned constraint, an optimal agent will avoid $s^c$ since it is infeasible and follows the second shortest path. In contrast, if we apply the penalty, an optimal agent will follow the shortest path since the sum of the reward and penalty is $9 - \eta$, which is greater than $9 - 2\eta$ for the second shortest path. Hence, *constraints and penalties lead to different inductive biases for generalization in new domains.* In this particular example, the constraint is independent of the transition dynamics and the reward function, while the effect of the penalty depends on the transition dynamics and reward function. Hence, in new environments, the constraint will make $s^c$ infeasible no matter what the transition dynamics and rewards are while the penalty may or may not ensure that $s^c$ is avoided depending on the reward function.

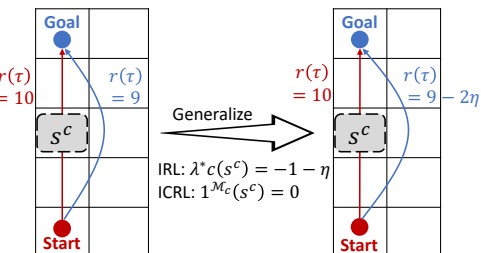

Figure 5: An example showing that generalizing $\tilde{r}$ and constraint $\mathbb{1}^{\mathcal{M}_c}(s^c) = 0$ learned in the training environment (Left) to new environment (Right) induce different optimal policies (the red path for $\tilde{r}$ and the blue for $\mathbb{1}^{\mathcal{M}_c}$).

Suppose we prefer a constraint for generalization purposes. It is possible to simply use an inverse RL technique to find $\bar{r}(s,a)$ and then convert it into a constraint function $c(s,a)$. Note that $\bar{r}(s,a) = r(s,a) + \lambda^* c(s,a)$. In order to recover $c(s,a)$, one needs to find the optimal Lagrange multiplier $\lambda^*$ of the Lagrangian dual, which is equivalent to solving the original constrained optimization problem. Hence, it is not clear that this will be simpler than directly learning a constraint. Nevertheless, this is a worthwhile direction for future work.

## 3.2 Constraint Inference in Inverse Optimal Control

Inverse Optimal Control (IOC) is a subfield that bridges the gap between machine learning and control theory. The primary objective of IOC problems is to deduce cost functions or constraint functions by closely observing expert behaviors. An IOC algorithm is comprised of a forward optimal control problem and a backward inference problem. More specifically, the forward problem can be characterized as follows:

$$\min_{\tau} f_0(\tau) \text{ s.t. } f_i(\tau) \leq \epsilon_{f,i}, \; h_i(\tau) = \epsilon_{h,i}, \; \forall i \in [1, I] \tag{14}$$

where function $f_0$ denotes the estimated cost (e.g., expectation or risk-sensitive metric) of the trajectories, while functions $f_i$ and $h_i$ establish the feasibility conditions through inequality and equality constraints. To deduce these constraints from the demonstration dataset $\mathcal{D}_E$, the backward inference problem can be

formulated as:

$$\text{find } f_i \text{ and } h_i \tag{15}$$
$$\text{s.t. } f_i(\tau_E) \leq \epsilon_{f,i}, h_i(\tau_E) = \epsilon_{h,i}, \ \forall \tau_E \in \mathcal{D}_E, \ \forall i \in [1, I],$$
$$f_i(\tau_{\neg *}) \geq \epsilon_{f,i}, h_i(\tau_{\neg *}) \neq \epsilon_{h,i}, \ \exists i \in [1, I] \tag{16}$$

where $\tau_{\neg *}$ denotes an unsafe trajectory. Intuitively, for any given expert trajectory $\tau_E \in \mathcal{D}_E$, it is necessary for the condition functions $f_i$ and $h_i$ to satisfy the constraints. Conversely, for every unsafe trajectory $\tau_{\neg *}$, these condition functions must lead to a violation of the constraints.

In this context, Chou et al. (2018) addressed the forward problem (as described in Equation 14) with the hit-and-run sampling algorithm (Kiatsupaibul et al., 2011). This algorithm generates lower-cost trajectories ($\tau_{\neg *}$) that comply with the constraints learned from the system. The design of sampling algorithms depends on the linearity and convexity of the underlying system dynamics. To solve the inverse problem, the state space is partitioned into discrete regions. A feasibility function is then learned to distinguish between feasible and infeasible states within these regions. Chou et al. (2019) and Chou et al. (2020) extended these algorithms to continuous state spaces. They achieved this by employing parametric constraint functions and devising uncertainty-aware constraints. These constraints were driven by robust optimization techniques and Bayesian inference, thereby enhancing the algorithms' adaptability and precision. A recent study (Papadimitriou & Li, 2023) proposed an incremental greedy constraint inference algorithm, which aims to minimize the KKT residual of the optimal control problem and infer constraints by progressively expanding the constraint set.

While the aforementioned studies have developed general constraint inference algorithms that can be extended to various tasks, a portion of IOC research specifically concentrates on applications that implement particular types of constraints. For instance, some research focuses on geometric constraints (Armesto et al., 2017; Pérez-D'Arpino & Shah, 2017), isoperimetric constraints (Wei et al., 2024b), and trajectory-oriented constraints (Li & Berenson, 2016; Mehr et al., 2016).

**IOC versus ICRL:** Although Constraint Inference IOC (CI-IOC) and ICRL address similar problems, their approaches to solving these problems are significantly different. The solution to IOC problems typically depends on the physical mechanism at play within a dynamical system, for example, the task of balancing a pendulum is commonly formulated as a linear quadratic problem. In this scenario, future states can be depicted as a linear mapping of prior states, with the primary objective being to develop a strategy that minimizes the quadratic cost. In this context, IOC aims to construct a closed-form solution by leveraging the known system dynamics or empirically estimating the model dynamics (i.e., model-based methods). In comparison, ICRL offers greater flexibility as it can not only incorporate model-based approaches to utilize system dynamics when available, but also leverage recent advancements in RL to learn a model-free controller without knowing or estimating the model dynamics (i.e., model-free methods). These distinctions underlie the fundamental differences between IOC and ICRL. In this study, we primarily focus on the formulation of ICRL problems and their potential solutions.

In the following sections, as illustrated in Figure 6, we introduce the main methodologies of ICRL in a structured manner.

## 4 Constraint Inference in Deterministic Environments

Under a deterministic environment, the environmental dynamics, such as transition and reward, follow a deterministic mapping. To conduct constraint inference under environments with deterministic dynamics, the Maximum Entropy (MEnt) RL approach is among the most extensively studied ICRL methods. In the rest of this section, we will introduce ICRL in both discrete and continuous domains under the MEnt framework.

### 4.1 Maximum Entropy ICRL in the Discrete Domain

Under the environment with deterministic dynamics, we start by considering the discrete state-action space. In this discrete domain, the goal of ICRL is to determine the most plausible set of constraints, denoted as

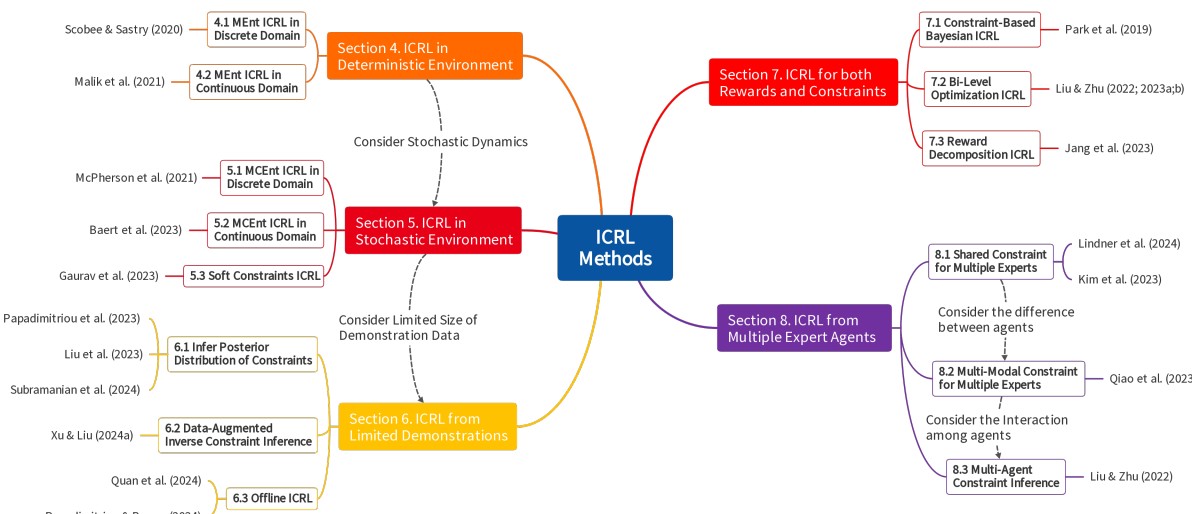

Figure 6: Illustrating the structure of ICRL methods. We describe the sections of our survey, including their relations and key literature. Specifically, we begin with the simplest ICRL setting in Section 4, where we assume deterministic dynamics. Over time, we gradually generalize to more realistic settings, beginning with stochastic dynamics (Section 5), then addressing demonstration data of limited size (Section 6), unknown reward functions (Section 7), and handling scenarios involving multiple agents (Section 8).

$C^*$, that can be incorporated into the original MDP $\mathcal{M}$ to explain the expert demonstrations $\mathcal{D}_E$. To obtain the optimal policy representation under the maximum entropy principle, Scobee & Sastry (2020) assume the expert demonstrations follow the underlying constraints, which are defined by identifiers $\mathbb{1}^{\mathcal{M}_c}(\tau_E) = 1$. The derived probability of observing trajectories within the dataset $\mathcal{D}_E$ can be represented as:

$$p_{\mathcal{M}_c}(\mathcal{D}_E|C) = \frac{1}{Z_c^{|\mathcal{D}_E|}} \prod_{\tau_E \in \mathcal{D}_E} e^{r(\tau_E)} \mathbb{1}^{\mathcal{M}_c}(\tau_E) \tag{17}$$

where $|\mathcal{D}_E|$ represents the size of the expert dataset, and $\mathcal{M}_c$ denotes the modified MDP after incorporating the inferred constraints. By employing the maximum likelihood method, the optimal constraint can be learned as follows:

$$C^* = \arg\max_{C \in \mathcal{C}} p_{\mathcal{M}_c}(\mathcal{D}_E|C) \tag{18}$$

Assuming the expert demonstrations are valid (i.e., $\mathbb{1}^{\mathcal{M}_c}(\tau_E) = 1$) and the rewards are known, the numerator in the likelihood $p_{\mathcal{M}_c}(\mathcal{D}_E|C)$ is independent of the learned policy. Maximizing $p_{\mathcal{M}_c}(\mathcal{D})$ can be equivalently modeled as minimizing the denominator $Z_c = \sum_{\tau \in \Xi_\mathcal{M}} e^{r(\tau)} \mathbb{1}^{\mathcal{M}_c}(\tau)$. Intuitively, this is accomplished by setting $\mathbb{1}^{\mathcal{M}_c}(\tau)$ to 0 for the trajectories with a large exponential reward $e^{r(\tau)}$ (i.e., a large generating probability) and ensuring these trajectories do not overlap with $\mathcal{D}_E$:

$$C^* = \arg\max_{C \in \mathcal{C}} \log p_{\mathcal{M}}(\Xi_{\overline{\mathcal{M}_c}}^-) - \alpha|C| \tag{19}$$
$$\text{s.t.} \quad \mathcal{D} \cap \Xi_{\overline{\mathcal{M}_c}}^- = \emptyset$$

where $\Xi_{\overline{\mathcal{M}_c}}^- = \{\tau \in \Xi_{\overline{\mathcal{M}_c}}^- | \mathbb{1}^{\mathcal{M}_c}(\tau) = 0\}$ represents the set of trajectories rendered infeasible by the added constraints, $|C|$ represents the size of the constraint set, and $\alpha$ serves as a hyperparameter that balances the trade-off between maximizing the probability and minimizing the constraint size. Intuitively, in contrast to incorporating all state-action pairs from non-expert trajectories into the constraint set, this objective effectively nullifies those non-expert trajectories that possess a large exponential reward $e^{r(\tau)}$. In terms of practical implementation, according to the objective in Equation (19), Scobee & Sastry (2020) devised a

constraint set by implementing a greedy iterative constraint inference algorithm, which iteratively adds a state into the constraint set.

## 4.2 Maximum Entropy ICRL in the Continuous Domain

Despite the convenience of discrete domains, in practice, underlying states and actions are often represented by continuous features, such as image patterns or text embeddings. This necessitates the modeling of continuous domains. In continuous state and action spaces, constructing the constraint set $\mathcal{C}^*$ is computationally intractable. Therefore, Malik et al. (2021) proposed learning a binary classifier $\phi_\omega(s, a)$ as a feasibility function approximator to determine the probability that performing action $a$ under state $s$ is feasible, such that $\mathbb{1}^{\mathcal{M}_c}(\tau) = \prod_{(s,a)\in\tau} \phi_\omega(s, a)$ denotes the probability that the trajectory $\tau$ is safe, and the corresponding cost can be defined as $c_\omega(\tau) = -\sum_{(s,a)\in\tau} \log \phi_\omega(s, a)$. Note that, compared to previous works Chou et al. (2019) that establish feasibility solely based on states, defining feasibility on both state and action pairs allows for modeling constraints that are inherently dependent on the state-action context.

In deterministic environments, previous ICRL works, including (Scobee & Sastry, 2020; Malik et al., 2021; Liu et al., 2023), often model hard constraints ($\epsilon = 0$). By employing the Karush-Kuhn-Tucker (KKT) condition and the interior point method with log barrier parameterized by $\beta$, we can derive the optimal policy and represent the probability of generating a trajectory $\tau$ as follows:

$$p_{\mathcal{M}_c}(\tau) = \frac{1}{Z_{c_\omega}} e^{r(\tau) + \beta \log[\prod_{(s,a)\in\tau} \phi_\omega(s,a)]} \tag{20}$$

where 1) $\beta$ balances the reward maximization and cost minimization in the objective and 2) the partition function can be denoted as $Z_{c_\omega} = \int_\tau e^{r(\tau) + \beta \log[\prod_{(s,a)\in\tau} \phi(s,a)]} d\tau$. Using this policy representation, the feasibility function $\phi_\omega$ can be adjusted to maximize the log-likelihood of generating expert data, i.e., $\max_\omega \log p_{\mathcal{M}_c}(\mathcal{D}_E)$. Taking the gradient of Equation (20) with respect to $\omega$ gives us:

$$\nabla_\omega \log p_{\mathcal{M}_c}(\mathcal{D}_E) = \nabla_\omega \log \prod_{\tau_E \in \mathcal{D}_E} p_{\mathcal{M}_c}(\tau_E) \tag{21}$$

$$= \sum_{\tau_E \in \mathcal{D}_E} \left[ \beta \nabla_\omega \log \left( \prod_{(s_E, a_E) \in \tau_E} \phi_\omega(s_E, a_E) \right) \right] - \mathbb{E}_{\tau \sim \pi(\tau)} \left[ \beta \nabla_\omega \log \left( \prod_{(s,a) \in \tau} \phi_\omega(s, a) \right) \right]$$

Similarly to constraint inference in a discrete environment, constraining all the states not covered by expert demonstrations is a trivial solution. To find the most effective constraint, Malik et al. (2021) incorporated the regularizer on the sparsity of cost into the objective function, and the final objective becomes:

$$\omega^* = \arg\max_\omega p_{\mathcal{M}_c}(\mathcal{D}_E) + \mathbb{E}_{\hat{\tau} \sim (\pi, \mathcal{D}_E)} |1 - \prod_{(s,a) \in \hat{\tau}} \phi_\omega(s, a)| \tag{22}$$

In order to construct a precise feasibility function within high-dimensional spaces, Malik et al. (2021) parameterized $\phi_\omega(\cdot)$ using neural networks and updated its parameters in accordance with the aforementioned objective.

Intuitively, the maximum likelihood loss classifies policy-visited states as infeasible and demonstration-visited states as feasible, but overlap can cause conflicting updates, slowing training and misclassifying safe states. To address this, Peng & Billard (2024) proposed a two-step Positive-Unlabeled Constraint Learning (PUCL) method, inspired by Positive-Unlabeled learning (Bekker & Davis, 2020), which first identifies reliable infeasible data and then trains a binary feasibility classifier as a constraint function using both positive and reliable infeasible data.

## 5 Constraint Inference in Stochastic Environments

Due to the complexity of real-world applications, deterministically predicting future states is challenging. In these applications, the environment dynamics, such as transition and reward functions, are often represented by stochastic models. However, the MEnt-based algorithms (Section 4) assume deterministic training

environments, without considering the influence of underlying stochasticity in the environment. Specifically, *stochastic transition functions introduce aleatoric uncertainty.* Striving for reliable constraint inference, the policy representation (e.g., the maximum entropy policy in 17) and the constraint model (e.g., $\phi_\omega$) must be sensitive to its influence.

In this section, in order to develop ICRL algorithms that are robust to the underlying uncertainty in the environment, we provide an overview of 1) Maximum Causal Entropy ICRL in both discrete (Section 5.1) and continuous (Section 5.2) domains, and 2) soft constraints (Section 5.3) that are compatible with the noise induced by stochastic dynamics.

### 5.1 Maximum Causal Entropy ICRL in the Discrete Domain

Similarly, we start by discussing the ICRL algorithm under the discrete domain. In the MDP with stochastic dynamics, the trajectory-level policy can be factorized into:

$$\pi(\tau) = \mu_0(s_0) \prod_{t=0}^{T} \pi(a_t|s_t) p_{\mathcal{T}}(s_{t+1}|s_t, a_t) \tag{23}$$

Consequently, modeling the trajectory-level policy (formula 17) requires accounting for the influence of transition dynamics $p_{\mathcal{T}}$, which is often unavailable to the agents during training (typically in the model-free setting). An effective approach that can generalize MEnt ICRL to stochastic environments is modeling the causal entropy (McPherson et al., 2021) framework. The causal entropy can be represented as:

$$\mathcal{H}(\boldsymbol{a}_{0:T}|\boldsymbol{s}_{0:T}) = \mathbb{E}_{\pi, p_{\mathcal{T}}, \mu_0}[\sum_{t=0}^{T} -\gamma^t \log \pi(a_t|s_t)] \tag{24}$$

where the step-wise policy $\pi(a|s)$ depends only on the available information at each step.

A critical challenge to constraint inference in stochastic environments lies in the definition of the feasibility function. In this discrete environment, McPherson et al. (2021) define:

$$\mathbb{1}^{\mathcal{M}_c}(s, a) = \mathbb{1}[p(S_{t+1} = \bar{s}|S_t = s, A_t = a) \leq \psi(\bar{s}), \forall \bar{s} \in C] \tag{25}$$

Intuitively, it assesses the likelihood that executing an action $a$ under a given state $s$ will result in an unsafe state $\bar{s}$ in the constraint set $\mathcal{C}$, and examine whether this likelihood is less than $\psi(\bar{s})$.

By following (Ziebart et al., 2010; Haarnoja et al., 2017), the optimal policy representation for Maximum Causal Entropy (MCEnt) Reinforcement Learning under feasibility function $\mathbb{1}^{\mathcal{M}_c}$ can be represented as follows :

$$\pi_{\mathcal{M}_c}(a_t|s_t) = \frac{e^{Q_{c,t}^{soft}(s_t, a_t)}}{e^{V_{c,t}^{soft}(s_t)}} \mathbb{1}^{\mathcal{M}_c}(s_t, a_t), \tag{26}$$

$$Q_{c,t}^{soft}(s_t, a_t) = r(s_t, a_t) + \mathbb{E}_{S_{t+1}}[V_{c,t+1}^{soft}(s_{t+1})], \tag{27}$$

$$V_{c,t}^{soft}(s_t) = \log \sum_{a_t} \mathbb{1}^{\mathcal{M}_c}(s_t, a_t) e^{Q_{c,t}^{soft}(s_t, a_t)} \tag{28}$$

In practical applications, these values can be calculated using soft Value Iteration, which is similar to conventional value iteration but replaces the hard maximum over actions with a log-sum-exp operation. By employing the aforementioned policy representation, the joint distribution within a horizon $[t : T]$ can be expressed as:

$$P_{\mathcal{M}_c}(A_{[t:T]} = a_{[t:T]}|S_t = s_t)$$
$$= \begin{cases} \frac{e^{\mathbb{E}[\sum_{\iota=t}^{T} r(s_\iota, a_\iota)]}}{e^{V_{c,t}^{soft}(s_t)}}, & \prod_{\iota=t}^{T}[\mathbb{1}^{\mathcal{M}_c}(s_\iota, a_\iota)] = 1 \\ 0, & \text{otherwise.} \end{cases} \tag{29}$$

From the aforementioned formula, it becomes evident that altering the constraint set $C$ only modifies the normalizing constant $e^{V_{c,t}^{soft}(s_t)}$. In accordance with the value function representation (Equation 26), expanding the constraint set results in a strict decrease in $V_{c,t}^{soft}(s_t)$, subsequently maximizing the likelihood of observed demonstrations. By utilizing the maximum likelihood estimation, the problem involves incrementally expanding the constraint set $C_0$ by iteratively adding $\bar{s}$ (i.e., $C_0 = C_0 \cup \bar{s}$) and determining the threshold $\psi(\bar{s})$ using the chance level specification algorithm (Vazquez-Chanlatte et al., 2018) for constructing $\mathbb{1}^{\mathcal{M}_c}(\cdot)$.

## 5.2 Maximum Causal Entropy ICRL in the Continuous Domain

In the continuous environments, constructing the constraint set is computationally intractable, so similar to Section 4.2, Baert et al. (2023) utilizes $\phi_\omega(s, a)$ to determine the probability that performing action $a$ under state $s$ is feasible. In this way, the constraint in the MCEnt objective can be updated to [2]:

$$-\mathbb{E}_{\tau \sim (\pi, \mathcal{T})}\Big[\sum_{t=0}^{T-1} \log \phi_\omega(s, a)\Big] \leq \epsilon \tag{30}$$

The policy satisfying the KKT condition can be represented as follows:

$$\pi_{\mathcal{M}_c}(a_t | s_t) = \frac{e^{Q_{c,t}^{soft}(s_t, a_t)}}{e^{V_{c,t}^{soft}(s_t)}} \tag{31}$$

$$Q_{c,t}^{soft}(s_t, a_t) = r(s_t, a_t) + \lambda \log \phi(s_t, a_t) + \mathbb{E}_{S_{t+1}}[V_{c,t+1}^{soft}(s_{t+1})] \tag{32}$$

$$V_{c,t}^{soft}(s_t) = \log \int_a e^{Q_{c,t}^{soft}(s_t, a)} \mathrm{d}a \tag{33}$$

In practice, these value functions can be effectively approximated using Soft Actor-Critic (Haarnoja et al., 2018) for continuous action spaces. To develop the maximum likelihood approach for updating the parameters of cost functions, Gleave & Toyer (2022) introduced the concept of the discounted likelihood of a trajectory:

**Definition 5.1** *(Discounted likelihood, Definition 3.1 in (Gleave & Toyer, 2022)). The discounted likelihood of a trajectory $\tau = (s_0, a_0, s_1, a_t, \ldots, a_{T-1}, s_T)$ under policy $\pi$ is:*

$$p_{\mathcal{M}_c}^\gamma(\tau) = \mu_0(s_0) \prod_{t=0}^{T-1} p_{\mathcal{T}}(s_{t+1} | s_t, a_t) \pi_{\mathcal{M}_c}(a_t | s_t)^{\gamma^t} \tag{34}$$

Intuitively, when $\gamma = 1$, the discounted likelihood is equivalent to the likelihood of $\tau$ under the policy $\pi$. For $\gamma \leq 1$, the probabilities of actions later in the trajectory are regularized by the power of $\gamma^t$. This regularization is useful in cases with an infinite planning horizon ($T \to \infty$), as it prevents the trajectory likelihood for any policy from approaching 0. Based on the aforementioned definition, the log-likelihood of a demonstrator's trajectories, sampled from demonstration distribution $\mathcal{D}_E$, is then:

$$\log p_{\mathcal{M}_c}^\gamma(\tau) = \sum_{t=0}^{T-1} \gamma^t \log \pi_{\mathcal{M}_c}(a_t | s_t) + \log \mu_0(s_0) + \sum_{t=1}^{T} \log p_{\mathcal{T}}(s_t | s_{t-1}, a_t) \tag{35}$$

$$= \sum_{t=0}^{T-1} \Big(Q_{c,t}^{soft}(s_t, a_t) - V_{c,t}^{soft}(s_t)\Big) + \log \mu_0(s_0) + \sum_{t=1}^{T} \log p_{\mathcal{T}}(s_t | s_{t-1}, a_t) \tag{36}$$

Based on the above formula, recent works (Gleave & Toyer, 2022; Baert et al., 2023) showed that the parameters of the constraints, denoted as $\omega$, can be updated to maximize the log-likelihood term $\log p_{\mathcal{M}_c}^\gamma(\tau)$ by utilizing the following gradient estimate:

$$\nabla_\omega \log p_{\mathcal{M}_c}^\gamma(\tau) = \mathbb{E}_{\mathcal{D}}\Big[\sum_{t=0}^{T-1} \gamma^t \nabla_\omega \phi_\omega(s_t, a_t)\Big] - \mathbb{E}_\pi\Big[\sum_{t=0}^{T-1} \gamma^t \nabla_\omega \phi_\omega(s_t, a_t)\Big] \tag{37}$$

---

[2] Baert et al. (2023) defines $\phi(\tau) = \sum_{t=0}^{T-1} \gamma^t \phi(s, a)$, but in the we denote $\phi(\tau) = \prod_{t=0}^{T-1} \phi(s, a)^{\gamma^t}$ so that $\phi(\tau)$ can represent the feasibility probability of a trajectory and the cost $c(\tau) = -\sum_t \gamma^t \log \phi(s_t, a_t)$.

## 5.3 Inferring Soft Constraints with ICRL

Unlike the hard constraints that guarantee constraint satisfaction, soft constraints can account for noise in sensor measurements, such as those caused by stochastic transition functions ($p_{\mathcal{T}}$) or cost functions ($p_{\mathcal{C}}$). This consideration helps address potential violations that may arise in expert demonstrations due to the stochastic dynamics in the environment. Specifically, the soft constraint can be represented as $\mathbb{E}_{p_{\mathcal{T}},\mu_0,\pi}\big[\sum_{t=0}^{T}\gamma^t c(s_t,a_t)\big] \leq \epsilon$ which is akin to the cumulative constraint (Eq. 2), but the soft constraints require a threshold $\epsilon > 0$. To better align with the soft constraint inference, the Inverse Soft Constraint Learning (ISCL) algorithm taken by (Gaurav et al., 2023) employs the Deep Constraint Correction (DC$^3$) framework (Donti et al., 2021), which transforms a constrained problem into an unconstrained problem by introducing a non-differentiable ReLU term. To extend this approach to ICRL, Gaurav et al. (2023) represented the CRL objective as:

$$\arg\max_{\pi} \mathbb{E}_{p_{\mathcal{T}},\mu_0,\pi}\Big[ \sum_{t=0}^{T}\gamma^t r(s_t,a_t)\Big] + \lambda\mathrm{ReLU}\Big[\mathbb{E}_{p_{\mathcal{T}},\mu_0,\pi}\Big(\sum_{t=0}^{T}\gamma^t c(s_t,a_t)\Big) - \epsilon\Big] \tag{38}$$

Upon obtaining the optimal policy $\pi$ during a specific run, ISCL incorporates it into the candidate policy set $\Pi$ such that $\Pi = \Pi \cup \pi$. Accordingly, the ICI objective in ICRL can be expressed as:

$$\arg\min_{c} -\mathbb{E}_{p_{\mathcal{T}},\mu_0,\pi_{mix}}\Big[ \sum_{t=0}^{T}\gamma^t c(s_t,a_t)\Big] + \lambda\mathrm{ReLU}\Big[\mathbb{E}_{p_{\tau\sim\mathcal{D}_E}}\Big(\sum_{t=0}^{T}\gamma^t c(s_t,a_t)\Big) - \epsilon\Big] \tag{39}$$

It is important to note that the first expectation is taken over the mixture policies $\pi_{mix}$, while the second expectation is derived from the expert dataset $\mathcal{D}_E$. ISCL constructs $\pi_{mix}$ as a weighted combination of the candidate policies in $\Pi$.

## 6 Constraint Inference from Limited Demonstrations

ICRL algorithms typically infer constraints from expert demonstrations, which necessarily have limited coverage of the underlying environment. Upon updating the constraint model, due to the limited amount of training data, epistemic uncertainty arises in game states that fall outside the data distribution. To be more specific, in an ICRL task, the training dataset $\mathcal{D}_{train}$ for constraint inference records the expert demonstrations and nominal trajectories generated by the imitation policy, i.e., $\mathcal{D} = \{\tau_{1,E},...,\tau_{N,E},\hat{\tau}_1,...,\hat{\tau}_N\}$. Epistemic uncertainty arises when the constraint model is asked to predict the costs of state-action pairs $(\bar{s},\bar{a})$ that are out of the training data distribution. This issue is closely related to the false correlation problem in offline RL (Jin et al., 2021; Xie et al., 2021), which is due to insufficient data coverage, leading to overestimated action values and suboptimal policies. Approaches to mitigate this challenge include conservative value estimation (Kumar et al., 2020), uncertainty modeling (Deng et al., 2023), and policy constraints (Wu et al., 2022). In the realm of ICRL, the algorithms discussed earlier primarily concentrate on addressing aleatoric uncertainty induced by the stochastic transition dynamics in the environment. Addressing the influence of epistemic uncertainty remains a critical problem in ICRL literature.

In this section, we provide an overview of ICRL algorithms that account for epistemic uncertainty, including the approach to estimating the posterior distribution of constraints (Section 6.1), data-augmented constraint inference (Section 6.2), and offline constraint inference (Section 6.3).

### 6.1 Modeling the Posterior Distribution of Constraints

The aforementioned ICRL methods typically learn a constraint function that best differentiates expert trajectories from generated ones. To better handle the limited training data, in this section, we introduce the ICRL method that models the posterior distribution of constraints.

**Bayesian Posterior Estimation.** In contrast to these traditional methods that primarily rely on maximum likelihood estimation, Papadimitriou et al. (2023) applied the Maximum-A-Posteriori (MAP) approach

to address the epistemic uncertainty during constraint inference. This led to the development of the Bayesian Inverse Constraint Reinforcement Learning (BICRL) model that infers a posterior probability distribution over constraints based on demonstrated trajectories.

BICRL is primarily designed to infer the constraint set[3] $C \in \{0,1\}^{|\mathcal{S}|}$ in the discrete state space $\mathcal{S}$. This is achieved by sampling candidate solutions, consisting of a candidate constraint set $\hat{C}$ and a Lagrange multiplier $\lambda$, from their inferred distributions at the previous run. Within the CMDP $\mathcal{M}_{\hat{C}}$ constructed on the candidate constraint set $\hat{C}$, BICRL computes the likelihood of expert demonstration $\tau_E$ under the MCEnt framework (see Section 5.1), so:

$$p_{\mathcal{M}_{\hat{C}},\lambda}(\tau_E|\hat{C},\lambda) = \mu_0(s_0) \prod_{t=0}^{T-1} p_{\mathcal{T}}(s_{t+1}|s_t, a_t) \prod_{t=0}^{T-1} \pi_{\mathcal{M}_{\hat{C}},\lambda}(a_t|s_t) = constant \cdot \prod_{t=0}^{T-1} \frac{e^{Q_{\hat{C},\lambda,t}^{soft}(s_t,a_t)}}{e^{V_{\hat{C},\lambda,t}^{soft}(s_t)}} \tag{40}$$

where the value functions can be defined by:

$$Q_{\hat{C},\lambda,t}^{soft}(s_t, a_t) = r(s_t, a_t) + \lambda \log \mathbb{1}^{\mathcal{M}_{\hat{C},t}}(s_t, a_t) + \mathbb{E}_{S_{t+1}} V_{\hat{C},t+1}^{soft}(s_{t+1}) \tag{41}$$

$$V_{\hat{C},t}^{soft}(s_t) = \log \int_a e^{Q_{\hat{C},t}^{soft}(s_t,a)} \mathrm{d}a \tag{42}$$

The feasibility identifier $\mathbb{1}^{\mathcal{M}_{\hat{C},t}}(s_t, a_t)$ defines whether performing the action $a_t$ in a state $s_t$ will lead to the transition to a constrained state $s_{t+1} \in \hat{C}$ (also see Equation (25)) and these (action)-value functions can be computed by dynamic programming (Sutton & Barto, 2018).

Within the $m$ runs, BICRL utilizes the learned distribution over constraint sets at previous $(m-1)^{th}$ as prior (i.e., $p_m(C,\lambda) = p_{m-1}(C,\lambda|\tau_E)$). The resulting posterior distribution can be represented as:

$$p_m(C,\lambda|\tau_E) = \frac{p_{\mathcal{M}_C,\lambda}(\tau_E|C,\lambda)p_m(C,\lambda)}{p(\tau_E)} \tag{43}$$

To initialize $p_0(C,\lambda)$, Papadimitriou et al. (2023) selected an uninformative prior. By utilizing the above posterior, the Maximum a Posteriori (MAP) estimates for the constraint sets and the penalty reward can be obtained as:

$$C_{MAP}, \lambda_{MAP} = \arg\max_{C,\lambda} p(C,\lambda|\tau) \tag{44}$$

and the Expected a Posteriori (EAP) estimates can be obtained as:

$$C_{EAP}, \lambda_{EAP} = \mathbb{E}_{C,\lambda \sim p(C,\lambda|\tau)}[C,\lambda|\tau] \tag{45}$$

Since sampling the constraint set $C$ from a continuous state space is computationally intractable, BICRL is mainly validated with discrete state spaces. How to extend this algorithm to continuous domains remains an open problem that requires further exploration.

**Variational Inference.** The aforementioned Bayesian updates depend on the samples derived from the Monte-Carlo Markov Chain (MCMC). However, this method tends to encounter intractability issues in environments with continuous state and action spaces. To tackle these problems, Liu et al. (2023) proposed Variational Inverse Constrained Reinforcement Learning (VICRL), which infers the approximated posterior distributions of constraints to capture uncertainty in the demonstration dataset.

Specifically, VICRL infers the distribution of a feasibility variable $\Phi$, such that $p_\omega(\phi|s,a)$ measures the extent to which an action $a$ should be allowed in a specific state $s$. The instance $\phi$ can define a soft constraint given by: $\hat{c}_\phi(s,a) = 1 - \phi$, where $\phi \sim p(\cdot|s,a)$. As $\Phi$ is a continuous variable within the range $[0,1]$, Liu et al. (2023) parameterized $p(\phi|s,a)$ using a Beta distribution:

$$\phi(s,a) \sim p_\omega(\phi|s,a) = \text{Beta}(\alpha_\omega, \beta_\omega) \text{ where } [\alpha_\omega, \beta_\omega] = \log[1 + \exp(f_\omega(s,a))] \tag{46}$$

---

[3]BICRL assumes a discrete state space, and $|\cdot|$ refers to the cardinality of a finite set.

Here $f_\omega$ is implemented by a multi-layer network with 2-dimensional outputs (for $\alpha$ and $\beta$). In practice, the true posterior $p(\phi|\mathcal{D}_E)$ is intractable for high-dimensional input spaces, so ICRL learns an approximate posterior $q(\phi|\mathcal{D}_E)$ by minimizing $\mathcal{D}_{kl}\Big[q(\phi|\mathcal{D}_E)\|p(\phi|\mathcal{D}_E)\Big]$. This is equivalent to maximizing an Evidence Lower Bound (ELBo):

$$\mathbb{E}_q\Big[\log p_{\mathcal{M}_c}(\mathcal{D}_E|\phi)\Big] - \mathcal{D}_{kl}\Big[q(\phi|\mathcal{D}_E)\|p(\phi)\Big] \tag{47}$$

In this case, the log-likelihood term $\log p_{\mathcal{M}_c}(\mathcal{D}_E|\phi)$ can be implemented using the trajectory likelihood (Eq. 20) within the MEnt framework, and the discounted likelihood (Eq. 34) within the MCEnt framework. The primary challenge in VICRL is defining the KL divergence. Aiming for ease in computing mini-batch gradients, (Liu et al., 2023) approximated $\mathcal{D}_{kl}\Big[q(\phi|\mathcal{D})\|p(\phi)\Big]$ with $\sum_{(s,a)\in\mathcal{D}}\mathcal{D}_{kl}\Big[q(\phi|s,a)\|p(\phi)\Big]$. As both the posterior and the prior follow Beta distributions, the KL divergence according to the Dirichlet VAE (Joo et al., 2020) can be represented as:

$$\begin{aligned}
\mathcal{D}_{kl}\Big[q(\phi|s,a)\|p(\phi)\Big] = &\log\Big(\frac{\Gamma(\alpha+\beta)}{\Gamma(\alpha^0+\beta^0)}\Big) + \log\Big(\frac{\Gamma(\alpha^0)\Gamma(\beta^0)}{\Gamma(\alpha)\Gamma(\beta)}\Big) \\
&+ (\alpha-\alpha^0)\Big[\psi(\alpha)-\psi(\alpha+\beta)\Big] + (\beta-\beta^0)\Big[\psi(\beta)-\psi(\alpha+\beta)\Big]
\end{aligned} \tag{48}$$

where 1) $[\alpha^0,\beta^0]$ and $[\alpha,\beta]$ are parameters from the prior and 2) the posterior functions and $\Gamma$ and $\psi$ denote the gamma and the digamma functions. Note that the goal of ICRL is to infer the smallest constraint for explaining expert behaviors. While previous methods often use a regularizer $\mathbb{E}[1-\phi(\tau)]$ (Malik et al., 2021) for punishing the scale of constraints, this KL-divergence term extends it by further regularizing the variances of constraints.

**Confidence-Aware Constraint Inference.** The aforementioned methods typically sample cost functions or constraints from the estimated distribution. However, when dealing with epistemic uncertainty, a more ideal approach is to first assess the confidence level in the estimated constraints. By doing so, one can ensure that only those constraints that meet a desired confidence threshold are utilized, thereby enhancing the reliability of the constraints used in the model. To achieve this goal, Subramanian et al. (2024) introduced Confidence-Aware Inverse Constrained Reinforcement Learning (CA-ICRL), which incorporates a confidence level alongside a set of expert demonstrations. This approach outputs a constraint that is at least as restrictive as the true underlying constraint based on a desired confidence level. The parameters of the constraint models $\omega$ are given by:

$$\omega = \arg\max_\omega \left[\sum_{\tau\in\mathcal{D}_E} r(\tau) + \log F^{-1}_{\text{Beta}(\alpha_\omega,\beta_\omega|\tau)}(1-\xi) - \log Z_\omega\right] \tag{49}$$

where $\xi$ denotes the desired confidence level, $Z_\omega$ denotes the partition function (similar to the $Z_c$ in Equation 20) and $F^{-1}_P(\xi)$ denotes the quantile function (i.e., inverse cumulative distribution) of distribution $P$ at a threshold $\xi$. Similar to (Liu et al., 2023), Subramanian et al. (2024) utilized the Beta distribution for modeling the feasibility of trajectories, and the parameters $(\alpha_\omega,\beta_\omega)$ are computed by aggregating the point-wise influence signals from each state-action pair within the trajectory. This is done with a deep set network that has the property of producing higher $(\alpha_\omega,\beta_\omega)$ parameters as the number of expert demonstrations increases, effectively reducing the epistemic uncertainty. The confidence-aware estimate of the feasibility of a trajectory is denoted by:

$$\phi^*(\tau) = F^{-1}_{\text{Beta}(\alpha_\omega,\beta_\omega|\tau)}(1-\xi) \tag{50}$$

In settings where developers can gather more expert demonstrations on a need basis, this confidence-aware framework can also be used to determine when we can stop gathering expert demonstrations to ensure that a desirable level of performance is achieved with a desired confidence level.

## 6.2 Data-Augmented Inverse Constraint Inference

Apart from modeling the posterior distribution of constraints, an alternative approach to handling epistemic uncertainty involves augmenting the dataset. Epistemic uncertainty arises due to the limited training data and the model's lack of knowledge about Out-of-Distribution (OoD) data. An effective measure of epistemic uncertainty is $I(\omega; y|x, \mathcal{D})$ (Smith & Gal, 2018; van Amersfoort et al., 2020), which quantifies the amount of information gained by the model $\omega$ when it observes the true label $y$ for a given input $x$, i.e., the greater the uncertainty of the model regarding the data, the more additional information it can obtain once the true label $y$ is observed. Under the setting of ICRL, to reduce the epistemic uncertainty, Xu & Liu (2024a) added the regularizer $I(\omega; \phi|\bar{\tau}, \mathcal{D})$ into the ICI objective (Equation 21) to propose the following objective:

$$\mathbb{E}_{\tau_E \in \mathcal{D}_E}\Big[\sum_{t=0}^{T}\log[\phi_\omega(s_t^E, a_t^E)]\Big] - \mathbb{E}_{\hat{\tau} \sim \hat{\mathcal{D}}}\Big[\sum_{t=0}^{T}\log[\phi_\omega(\hat{s}_t, \hat{a}_t)]\Big] - \alpha I(\omega; \phi|\bar{\tau}, \mathcal{D}) \tag{51}$$

where $\mathcal{D}$ denotes the training dataset consisting of expert demonstration $\mathcal{D}_E$ and imitation demonstrations $\hat{\mathcal{D}}$. Since the mutual information term $I$ is computationally intractable, Xu & Liu (2024a) showed it can be empirically approximated by $I(\omega; \phi|\bar{\tau}, \mathcal{D}) = \mathcal{H}[p(\phi|\bar{\tau}, \mathcal{D})] - \frac{1}{M}\sum_m \mathcal{H}[p(\phi|\bar{\tau}; \omega_m)]$ where $\omega_m \sim q(\omega)$. Specifically, 1) $\mathcal{H}[p(\phi|\bar{\tau}, \omega_m)]$ defines the entropy of a constrained model parameterized by $\omega_m$. 2) $\mathcal{H}[p(\phi|\bar{\tau}, \mathcal{D})] \in [0, \infty)$ measures the amount of information required to describe the feasibility $\phi$ of an exploratory trajectory $\bar{\tau}$ based on the given training dataset $\mathcal{D}$. By substituting them in formula (51), we obtain the following objective:

$$\frac{1}{M}\sum_m \mathbb{E}_{\mathcal{D}_E}\Big[\sum_{t=0}^{T}\log[\phi_\omega(s_t^E, a_t^E)]\Big] - \mathbb{E}_{\hat{\mathcal{D}}}\Big[\sum_{t=0}^{T}\log[\phi_\omega(\hat{s}_t, \hat{a}_t)]\Big] - \alpha\mathcal{H}[p(\phi|\bar{\tau}, \mathcal{D})] + \alpha\mathcal{H}[p(\phi|\bar{\tau}; \omega_m)] \tag{52}$$

Inspired by (Smith & Gal, 2018), Xu & Liu (2024a) used dropout layers (Srivastava et al., 2014) for approximating the distribution of model parameters $q(\omega)$. Besides, to reduce the conditional entropy $\mathcal{H}[p(\phi|\bar{\tau}, \mathcal{D})]$, Xu & Liu (2024a) proposed expanding the training dataset by adding generated trajectories $\{(\tau_G, \phi_G)\}$. To be more specific, the augmented expert dataset is constructed by $\mathcal{D}_E^G = \mathcal{D}_E \cup \tau_G, \forall \phi_G = 1$, and the augmented nominal dataset is constructed by $\hat{\mathcal{D}}^G = \hat{\mathcal{D}} \cup \tau_G, \forall \phi_G = 0$. By substituting them in objective (52), we arrive at the following objective:

$$\frac{1}{M}\sum_m \mathbb{E}_{\mathcal{D}_E^G}\Big[\sum_{t=0}^{T}\log[\phi_{\omega_m}(s_t^E, a_t^E)]\Big] - \mathbb{E}_{\hat{\mathcal{D}}^G}\Big[\sum_{t=0}^{T}\log[\phi_{\omega_m}(\hat{s}_t, \hat{a}_t)]\Big] + \alpha\mathcal{H}[p(\phi|\bar{\tau}; \omega_m)] \tag{53}$$

In order to generate the trajectories $\tau_G$, Xu & Liu (2024a) designed a Flow-based Trajectory Generation (FTG) algorithm that extends the Generative Flow Network (GFlowNet) (Bengio et al., 2021) to generate a diverse set of trajectories based on the dataset and task-dependent rewards. Similar to the generative data synthesizer (Zhu et al., 2023), FTG explores various combinations of "points" (such as state-action pairs in RL) within sequential data. This design enables FTG to generate trajectories in which the sequence of states and actions deviates from those in the training data samples through multiple rounds of random sampling (Li et al., 2023). The dataset augmented with these trajectories can more accurately characterize the underlying distribution of feasible and infeasible trajectories, thereby reducing the uncertainty associated with the parameters of the constraint model.

## 6.3 Offline Inverse Constraint Inference

The aforementioned methods primarily focus on the impact of limited expert demonstrations. However, the imitation policy can be learned through interaction with the environment. In contrast, a more stringent scenario is Offline ICRL, where the agent must infer constraints and learn imitation policies based solely on a fixed dataset, without access to the environment for additional interactions.

Specifically, in Offline ICRL, we are given an offline dataset $\mathcal{D}_O = \{s_n^O, a_n^O, r_n^O\}_{n=1}^{N_O}$. Within this dataset, expert trajectories are denoted as $\mathcal{D}_E = \{s_n^E, a_n^E, r_n^E\}_{n=1}^{N_E} \subset \mathcal{D}_O$. These expert trajectories are generated by an optimal policy $\pi^E$ adhering to the unobserved constraint function $c(s, a)$. More formally, $\pi^E = \pi_c^* =$

$\arg\max_\pi \mathbb{E}_{\rho^{\pi^E}}[r(s,a)]$, subject to $\mathbb{E}_{\rho^{\pi^E}}[c(s,a)] \leq \epsilon$ ($\rho^{\pi^E}$ denotes the occupancy measure by following the expert policy $\pi^E$). An estimated cost function $\hat{c}$ is a feasible solution to ICRL if and only if the optimal solution $\pi_{\hat{c}}^*$ can reproduce $d^E$. To achieve offline ICRL, Quan et al. (2024) mainly study hard constraints with $\epsilon = 0$. The CRL problem can be formulated as:

$$\max_\pi \mathbb{E}_{\rho^\pi}[r(s,a)] \tag{54}$$
$$\text{s.t. } \rho^\pi(s,a)c(s,a) \leq 0, \quad \forall s,a$$

By extending this objective to the offline ICRL problem, this objective can be updated to:

$$\max_{\rho(s,a)c(s,a)\leq 0, \rho(s,a)\geq 0} \mathbb{E}_\rho[r(s,a)] - \beta_r \mathcal{D}_f(\rho\|\rho^O) \tag{55}$$
$$\text{s.t. } \sum_{a\in\mathcal{A}} \rho(s,a) = (1-\gamma)\rho_0(s) + \gamma \sum_{s'\in\mathcal{S},a'\in\mathcal{A}} \rho(s',a')p(s|s',a'), \ \forall s \in \mathcal{S}$$

where $\rho^O$ represents the visitation distribution in the offline dataset $\mathcal{D}^O$, $\mathcal{D}_f(\rho\|\rho^O)$ denotes the $f$-divergence between two distributions, and $\beta_r$ denotes the weighting parameter. Intuitively, instead of maximizing only the reward, we augment the cumulative reward with a divergence regularizer to prevent it from deviating beyond the coverage of the offline data. Note that this objective forms a dual problem for CRL (Sikchi et al., 2024).

To align with the above Distributional Correction Estimation (DICE) (Lee et al., 2021) objective, Quan et al. (2024) translate the constraint inference problem into the problem of estimating the *superior distribution set* as follows.

**Definition 6.1** *The set of superior distributions (the distribution is on state-action pairs. e.g., the normalized occupancy measure defined in Equation 4), denoted as $\mathfrak{O}$, is defined as those distributions that are generated by a certain policy $\pi$ and achieve higher cumulative rewards than experts, denoted as $\mathfrak{O} = \{\rho^\pi : \mathbb{E}_{\rho^\pi}[r(s,a)] > \mathbb{E}_{\rho^E}[r(s,a)]\}$.*

Intuitively, for any superior distribution $\rho^* \in \mathfrak{O}$, there's at least one state-action pair $(s,a)$ with $\rho^*(s,a) > 0$ that constitutes a violation of the constraints. A straightforward method to identify a feasible cost function $c$ begins with estimating the set of superior distributions $\mathfrak{O}$. Subsequently, a positive cost is assigned to at least one state-action pair within the support of each distribution $\rho^*$ included in this set. This assignment is done while ensuring that the chosen state-action pair is not covered by the expert demonstrations.

A recent study (Papadimitriou & Brown, 2024) explored an alternative scenario in which preferences among demonstrations are available. Specifically, Papadimitriou & Brown (2024) developed the Preference-Based Bayesian Inverse Constraint Reinforcement Learning (PBICRL) algorithm. This method extends the Bradley-Terry model to the context of constraint inference. In this approach, constraints are inferred by maximizing the likelihood that demonstrations with higher preferences ($\mathcal{D}_+$) are more effective than those with lower preferences ($\mathcal{D}_-$). This log-likelihood function is represented as follows:

$$\mathcal{L}(w,c) = \sum_{\tau_i\in\mathcal{D}_+,\tau_j\in\mathcal{D}_-} \log p(\tau_i > \tau_j) = \sum_{\tau_i\in\mathcal{D}_+,\tau_j\in\mathcal{D}_-} \log \frac{e^{\frac{\beta}{T_i}\sum_{s\in\tau_i}[r(s)+w\cdot c(s)]-m_{+-}}}{e^{\frac{\beta}{T_i}\sum_{s\in\tau_i}[r(s)+w\cdot c(s)]-m_{+-}} + e^{\frac{\beta}{T_j}\sum_{s\in\tau_j}[r(s)+w\cdot c(s)]}} \tag{56}$$

where 1) $w$ denotes the weighting term for the cost $c$, 2) $\beta$ denotes the inverse temperature parameter, 3) $T_i$ indicates the length of a trajectory $\tau_i$ and 4) $m_{+-}$ is the margin parameter between group $\mathcal{D}_+$ and $\mathcal{D}_-$.

## 7 Simultaneous Inference of Rewards and Constraints

Previous ICRL algorithms typically focus on learning a constraint function based on a known reward function. However, these methods struggle in environments where both constraint and reward signals are absent. Under this setting, an intriguing yet relatively unexplored extension of ICRL is to simultaneously infer rewards and constraints from expert demonstrations. However, since both IRL and ICRL are inherently ill-posed due

to the ambiguity in identifying the reward function and constraint function, concurrently inferring rewards and constraints can substantially amplify the complexity and ambiguity of identifying the true underlying rewards and constraints.

In this section, we provide an overview of algorithms that learn both rewards and constraints, including constraint-based Bayesian IRL (Park et al., 2019) (Section 7.1) that learns different local rewards and local constraints from different expert trajectory segments, maximum-likelihood ICRL (Liu & Zhu, 2022; 2023b;a) (Section 7.2) that learns global rewards and constraints from complete trajectories, and transferable constraint learning (Jang et al., 2023) that jointly infers task reward and residual task agnostic constraint pairs from demonstrations.

## 7.1 Constraint-Based Bayesian Inverse Reinforcement learning

While the existing IRL and ICRL methods typically learn a single reward function or a single constraint function from the expert trajectories, Park et al. (2019) proposed a Constraint-based Bayesian Nonparametric IRL (CBN-IRL) algorithm that learns multiple local rewards/goals and local constraints from different expert trajectory segments. Specifically, Park et al. (2019) splits the expert trajectory $\tau_E$ into multiple partitions $\{\iota_j\}_{j=1}^J$. Within each partition $\iota$, CBN-IRL learns a goal function $g_\iota(s)$ that assigns a positive reward to the destination $s_\iota^*$ and zero to other states, i.e., $g_\iota(s) = \mathbb{1}(s = s_\iota^*)$ where $s_\iota^*$ is the goal state. CBN-IRL also learns a constraint function $c_\iota(s) \to \{0, 1\}$ assigns a value of one to infeasible or unsafe states and zero to feasible or safe states. Following this setting, Park et al. (2019) proposed the Maximum-A-Posterior (MAP) objective to infer the constraint $c_\iota(s)$ and goal $g_\iota(s)$ for each segment.

$$(g^*, c^*) = \arg\max_{g,c} \prod_{j=1}^J p(\tau_{E,j}|g_{\iota_j}, c_{\iota_j}) p(\iota_j|\iota_{-j}) p(g_0, c_0) \tag{57}$$

where $p(\tau_{E,j}|g_{\iota_j}, c_{\iota_j})$ denotes the likelihood of generating the expert trajectory segment $\tau_{E,j}$ in the $j^{th}$ partition $\iota_j$, $p(\iota_j|\iota_{-j})$ refers to the probability of transferring from other partitions $\iota_{-j}$ to partition $\iota_j$, and $p(g_0, c_0)$ denotes the prior. Partitioning the expert trajectory into local segments increases the sparsity of rewards and reduces the complexity of constraint inference for each partition.

## 7.2 Bi-Level Optimization Inverse Constrained Reinforcement Learning

While CBN-IRL learns multiple local reward/goal functions and local constraint functions from different expert trajectory segments, Liu & Zhu (2022; 2023a;b) proposed to learn a single reward function and constraint function for the complete expert trajectories. Specifically, Liu & Zhu (2023a;b) formulated a bi-level optimization problem:

$$\max_r \ \mathbb{E}_{\tau \sim \mathcal{D}_E}\Big[ \sum_{t=0}^T \gamma^t \log \pi_{c^*(r);r}(a_t|s_t) \Big]$$

$$\text{s.t.} \quad c^*(r) := \arg\min_c \ \mathbb{E}_{p_\mathcal{T}, \pi_{c;r}, \mu_0}\Big[ \sum_{t=0}^T \gamma^t[r(s_t, a_t) - c(s_t, a_t)] \Big] + \mathcal{H}(\pi_{c;r}) + \mathbb{E}_{\tau \sim \mathcal{D}_E}\Big[ \sum_{t=0}^T \gamma^t c(a_t|s_t) \Big] \tag{58}$$

The upper level aims to learn a reward function $r$ to maximize the log-likelihood of the expert trajectories $\mathcal{D}_E$ where $\pi_{r;c}$ is the constrained soft Bellman policy Liu & Zhu (2022; 2023a;b) under the reward function $r$ and constraint function $c$. It is proven in Liu & Zhu (2022; 2023a) that the constrained soft Bellman policy maximizes the entropy-regularized cumulative reward-minus cost:

$$\pi_{r;c} = \arg\max_\pi \mathbb{E}_{p_\mathcal{T}, \pi, \mu_0}\Big[ \sum_{t=0}^T \gamma^t[r(s_t, a_t) - c(s_t, a_t)] \Big] + \mathcal{H}(\pi) \tag{59}$$

The lower-level objective function can be partitioned into two parts. The first part $\mathbb{E}_{p_\mathcal{T}, \pi_{c;r}, \mu_0}\Big[ \sum_{t=0}^T \gamma^t[r(s_t, a_t) - c(s_t, a_t)] \Big] + \mathcal{H}(\pi_{c;r})$ uses adversarial learning to encourage a constraint

function $c$ that makes the best policy $\pi_{r;c}$ perform the worst. The second part $\mathbb{E}_{\tau \sim \mathcal{D}_E}\left[\sum_{t=0}^{T} \gamma^t c(a_t|s_t)\right]$ penalizes constraint functions where the expert has high cumulative constraint.

However, this bi-level formulation (58) still suffers from unidentifiabilty, i.e., infinitely many combinations of reward and constraints can explain the expert trajectories. Therefore, Liu & Zhu (2022) assume that the reward function and constraint function are linearly parameterized, so that they can prove and leverage the strictly convex property of the lower-level problem to find a unique constraint function and thereby guarantee the convergence of the reward function.

### 7.3 Reward Decomposition Inverse Constrained Reinforcement Learning

Jang et al. (2023) proposed decomposing the reward function into *task* reward $r_p \in \mathcal{R}_p$ and the constraint-related *residual* reward $r_c \in \mathcal{R}$, given the constrained demonstrations $\mathcal{D}_E$ and the task reward space $\mathcal{R}_p \subseteq \mathcal{R}$. Here, $r_p$ is to produce an expert-like but unconstrained behavior, defined based on predefined task-relevant features. In contrast, $r_c$ represents a negative constraint-cost function, expressed as $r_c = -c(s, a)$. By designing a Transferable Constraint Learning (TCL) algorithm, the learned $r_c$ is task-agnostic and transferable across tasks to produce constrained behaviors.

Following the Q-decomposition framework (Russell & Zimdars, 2003), Jang et al. (2023) designed an optimization problem that simultaneously learns the overall reward $r$ from demonstrations $\mathcal{D}_E$ through inverse RL and decomposes $r$ into an optimal reward pair $(r_p^*, r_c^*)$ using a reward decomposition (RD) approach:

$$(r_p^*, r_c^*) = \arg \max_{r_p \in \mathcal{R}_p, r_c \in \mathcal{R}} \left(\min_{\pi \in \Pi} J_{\mathrm{IRL}}(r, \pi; \mathcal{D}_E) - J_{\mathrm{RD}}(r_p, \pi; \mathcal{R}_p)\right) \tag{60}$$
$$\text{s.t. } r = r_p + r_c$$

where $J_{\mathrm{IRL}}$ and $J_{\mathrm{RD}}$ are objective functions and $\pi$ is an output policy associated with the overall reward $r$.

Specifically, $J_{\mathrm{IRL}}$ is the maximum causal entropy IRL (Ziebart et al., 2010) objective function aiming to infer a total reward $r$ and its associated policy $\pi$ that can produce behaviors similar to demonstrations $\mathcal{D}_E$:

$$J_{\mathrm{IRL}}(r, \pi; \mathcal{D}_E) = \mathbb{E}_{s,a \sim \mathcal{D}_E}\left[r(s, a)\right] - \mathbb{E}_{s,a \sim \pi}\left[r(s, a)\right] - \mathcal{H}(\pi) \tag{61}$$

$J_{\mathrm{RD}}$ is the reward decomposition function designed to determine $r_c$ from the overall reward $r$, based on the assumption of an additive decomposition $r = r_p + r_c$. Jang et al. (2023) utilized $r_p$ to define a task-specific policy $\pi_p$, which governs all task-relevant but unconstrained actions. By identifying the largest $r_p$ that closely approximates $r$, the residual reward $r_c$ can be derived. This residual reward, being task-agnostic, enables its corresponding policy to generalize across different tasks. To achieve this, the objective function $J_{\mathrm{RD}}$ is formulated to minimize the action divergence between $\pi_p$ and $\pi$:

$$J_{\mathrm{RD}}(r_p, \pi; \mathcal{R}_p) = \mathbb{E}_{s \sim \rho_{\mathcal{M}}^{\pi}}\left[D_{\mathrm{KL}}\big(\pi_p(\cdot \mid s)\|\pi(\cdot \mid s)\big)\right] \tag{62}$$

where $\rho_{\mathcal{M}}^{\pi}$ is the state-visitation frequencies for $\pi$, $\pi_p$ define the policy learned under the $r_p$ and $D_{\mathrm{KL}}(\cdot)$ represents a Kullback-Leibler (KL) divergence between two given policies. In Jang et al. (2023), task-relevant features are manually selected to serve as the basis vectors for the task space. When such features are not readily available, alternative options include using features designed for sparse rewards or binary indicators, which can effectively represent the task structure.

## 8 Constraint Inference from Multiple Expert Agents

In practice, demonstration datasets may be generated by multiple agents. For instance, on an open road, vehicle trajectories could be produced by a variety of human drivers, each possessing different driving skills (e.g., risk-averse and risk-seeking) and operating different types of vehicles (i.e., trucks, vans, or cars). These environments require considering the behaviors of multiple heterogeneous and homogeneous agents.

Unlike Multi-Agent Reinforcement Learning (MARL) (Zhang et al., 2021), which solves only forward control problems, ICRL typically involves solving a backward inverse constraint inference problem using provided

expert demonstrations. To adapt ICRL to multi-agent settings, we define three separate levels of Multi-Agent ICRL:

1) Expert trajectories are generated by multiple types of agents, all adhering to the same constraint but optimizing different reward functions. The key challenge at this level is: *How can we infer the shared constraint respected by various types of agents?*

2) Expert trajectories are generated by multiple types of agents, each respecting different constraints. The key challenge lies in: *How can different constraints be inferred for various types of agents?*

3) The demonstration dataset is produced by multiple agents acting simultaneously. The challenge here is to determine: *How can cooperative and competitive behaviors among the agents be modeled in order to infer appropriate constraints for explaining their behaviors?*

To better differentiate these challenges, in the first two research topics, we assume the expert demonstrations are generated by multiple experts, but the forward control policy is conditioned on a single agent type by treating the states of other agents as background. However, for the final challenge, interactions among multiple agents must be considered to model the joint behavior of different agents. In this section, we study several existing ICRL algorithms that aim to tackle these challenges.

## 8.1 Inferring a Shared Constraint from Multiple Expert Demonstrations

In order to infer a consistent constraint respected by multiple types of expert agents, Lindner et al. (2024) studied a setting where the expert agents optimize different rewards under a shared constraint. To represent the constraint, Lindner et al. (2024) defined the safe set as the convex hull of the feature expectations of the expert demonstrations:

$$\neg \mathcal{C} = \text{conv}(\mathcal{D}_E) := \{\sum_k \lambda_k f(\pi_{E,k}) | \lambda_k \geq 0 \text{ and } \sum_k \lambda_k = 1\} \tag{63}$$

where $f(\pi_{E,k}) = \mathbb{E}_{\pi_{E,k}, p_{\mathcal{T}}}[\sum_{t=0}^T \gamma^t f(s_t, a_t)]$ estimates the feature expectations derived by the $k^{th}$ expert policy $\pi_{E,k}$. The inferred constraint essentially establishes a 'worst-case (pessimistic) constraint', as opposed to an indispensable one (i.e., the minimal constraint in Section 2.4). Under this inferred constraint, policies whose feature expectation is not in the convex hull represented by the weighted combination of expert policies' feature expectations are considered to be constraint-violating. Intuitively, a policy exploring the state-action pairs uncovered by (i.e., located outside the support of) expert demonstration is automatically considered infeasible.

In Lindner et al. (2024), instead of learning a constraint function, the approach assumes that any unseen behavior is unsafe. It enforces constraints by requiring the learner to act as a convex combination of the demonstrated safe trajectories. The key advantage of this method is that it eliminates the need to know the reward function that the expert was optimizing. However, by restricting the learner to merely replicate the expert's demonstrated behavior, this approach limits the ability to generalize effectively and may result in highly suboptimal performance on new tasks. To solve this problem, Kim et al. (2023) additionally leverages side information in the form of a reasonable set of constraints, enabling policy performance guarantees.

Specifically, Kim et al. (2023) considers inverse constraint learning under the assumption that the reward function, expert demonstrations, and a class of potential constraints $\mathcal{F}_c$ are available. It is assumed that the ground-truth constraint $c^*$ lies within the convex and compact set $\mathcal{F}_c$. Building upon this, Kim et al. (2023) formulates the problem of inferring a shared constraint from multi-task demonstrations as follows:

$$\max_{c \in \mathcal{F}_c} \min_{\pi^{1:K} \in \Pi} \max_{\lambda^{1:K} > 0} \sum_i^K \left( J(\pi_E^i, r^i - \lambda^i c) - J(\pi^i, r^i - \lambda^i c) \right) \tag{64}$$

where $J(\pi, f) = \mathbb{E}_{\tau \sim \pi} \left[ \sum_{t=0}^T f(s_t, a_t) \right]$ denotes the value of policy $\pi$ under reward or cost function $f$, based on the observed $K$ samples of the form $(r_k, \{\tau \sim \pi_E^k\})$.

## 8.2 Multi-Modal Constraint Inference from a Mixture of Expert Demonstrations

Instead of assuming the agents respect the same constraint, Qiao et al. (2023) studied expert data $\mathcal{D}_E$ that record demonstrations from multiple experts who respect different kinds of constraints. To infer these constraints from a mixture of expert demonstrations, (Qiao et al., 2023) proposed a Multi-Modal Inverse Constrained Reinforcement Learning (MM-ICRL) algorithm that performs unsupervised agent identification and multi-modal policy optimization to learn agent-specific constraints.

Specifically, MM-ICRL trained flow-based density estimator $p_\psi(s, a|k)$ based on Masked Auto-regressive Flow (MAF) (Papamakarios et al., 2017). This density estimator is trained to maximize the log-likelihood of trajectories generated by a specific agent $k$. The agent's trajectory-level identifier can be represented using the softmax representation such that $p_\psi(k|\tau) = \frac{\exp \prod_{(s,a)\in\tau} p_\psi(s,a|k)}{\sum_{k'} \exp \prod_{(s,a)\in\tau} p_\psi(s,a|k')}$. After learning the density model $p_\psi(s,a|k)$, MM-ICRL divides $\mathcal{D}_E$ into sub-datasets $\{\mathcal{D}_k\}_{k=1}^{|\mathcal{K}|}$ by: 1) initializing the dataset $\mathcal{D}_k = \emptyset$ and 2) $\forall \tau_i \in \mathcal{D}_E$, adding $\tau^i$ into $\mathcal{D}_k$ if $k = \arg\max_k p_\psi(k|\tau)$. We repeat the above steps for all $k \in \mathcal{K}$. Based on the identified expert dataset $\mathcal{D}_k$, Qiao et al. (2023) performs *agent-specific constraint inference* to learn the conditional feasibility function $\phi_\omega(s_t, a_t|k)$ and updated the parameters $\omega$ by computing the gradient of the conditional likelihood function:

$$\nabla_\omega \log\left[p(\mathcal{D}_k|\phi, k)\right] = \sum_{i=1}^{N}\left[\nabla_\omega \sum_{t=0}^{T} \eta \log[\phi_\omega(s_t^{(i)}, a_t^{(i)}|k)]\right] - N\mathbb{E}_{\hat{\tau}\sim\pi_{\mathcal{M}^\phi}(\cdot|k)}\left[\nabla_\omega \sum_{t=0}^{T} \eta \log[\phi_\omega(\hat{s}_t, \hat{a}_t|k)]\right] \quad (65)$$

This inverse constraint objective relies on the nominal trajectories $\hat{\tau}$ sampled with the conditional imitation policy $\pi_{\mathcal{M}^{\hat{\phi}}}(\tau|k)$. MM-ICRL learns the imitation policy by following the *multi-modal policy optimization* objective:

$$\min_\pi -\mathbb{E}_{\pi(\cdot|k)}\left[r(\tau) + \alpha_1 \log[p_\psi(k|\tau)]\right] + (\alpha_2 - \alpha_1)\mathcal{H}[\pi(\tau|k)] \quad (66)$$

$$\text{s.t. } \mathbb{E}_{\pi(\cdot|k)}\left(\sum_{t=0}^{h} \gamma^t \log \phi_\omega(s, a, k)\right) \geq \epsilon$$

Intuitively, this objective expands the reward signals with a log-probability term $\log[p_\psi(k|\tau)]$, which encourages the policy to generate trajectories from high-density regions conditioning on a specific agent type. This approach ensures that the learned policies $\pi(\cdot|k)_{k=1}^{K}$ are differentiable.

The MM-ICRL method alternates between executing agent-specific constraint inference and optimizing multi-modal policies. This process is accompanied by updating density models using the acquired limitation policies until they successfully reproduce the expert demonstration.

## 8.3 Inverse Constraint Inference from Multi-Agent Environment

For expanding ICRL to model the cooperative behaviors among multiple agents, one effective approach is to adopt a Constrained Markov Game (CMG) framework where the action space is $\mathcal{A} = \prod_{k=1}^{K} \mathcal{A}^{[k]}$, denoting that multiple agents perform simultaneously. In a CMG environment, Liu & Zhu (2022) considered a collaborative multi-agent setting, where multiple agents collaboratively recover the expert constraints. Assuming an additively decomposed cost function $c(s,a) = \sum_{k=1}^{K} c_k(s,a)$, the corresponding constraint can be represented by:

$$\mathbb{E}_{\pi, p_\mathcal{T}, \mu_0}\left[\sum_{t=0}^{T} \gamma^t \sum_{k=1}^{K} c_k(s,a)\right] \leq \epsilon \quad (67)$$

To find a statistical distribution model for the policy $\pi$ based on the above constraint, Liu & Zhu (2022) formulated the problem based on the MCEnt framework (see Section 5.1) and theoretically showed that the

optimal solution follows the constrained soft Bellman policy given by:

$$\pi_{\mathcal{M}_c} = \arg\max_{\pi} \mathbb{E}_{\pi, p_{\mathcal{T}}, \mu_0}\Big[\sum_{k=1}^{K} \lambda_{r,k} \sum_{t=1}^{T} \gamma^t f_k(s_t, a_t)\Big]$$

$$- \lambda_c \mathbb{E}_{\pi, p_{\mathcal{T}}, \mu_0}\Big[\sum_{t=0}^{T} \gamma^t \sum_{k=1}^{K} c_k(s, a_k)\Big] + \sum_{t=0}^{T} \mathbb{E}_{\pi, p_{\mathcal{T}}, \mu_0}\Big[\gamma^t \log \pi(a_1, \ldots, a_K | s_t)\Big], \tag{68}$$

where $f$ represents some fixed feature mapping for expert $k$, $\lambda_{r,k}$ and $\lambda_c$ are the optimal dual variables. Given the representation of the optimal multi-agent policy $\pi_{\mathcal{M}_c}$, the constraint is updated by maximizing the log-likelihood of generating the expert demonstrations by utilizing the optimal policy:

$$c^* = \arg\max_{c} \sum_{\tau_E \in \mathcal{D}_E} \sum_{t=0}^{T} \gamma^t \log[\pi_{\mathcal{M}_c}(a_{E,t}|s_{E,t})] \tag{69}$$

# 9 Benchmarks and Applications

In this section, we introduce the existing benchmarks for evaluating ICRL algorithms and explore the potential applications of ICRL in addressing critical real-world challenges.

## 9.1 Benchmarks

**Discrete Environments.** *Grid-World* environments are among the most well-studied discrete environments for evaluating the performance of ICRL algorithms. Specifically, previous works (Scobee & Sastry, 2020; McPherson et al., 2021; Papadimitriou et al., 2023; Glazier et al., 2021; Gaurav et al., 2023) added some obstacles to a grid map and examined whether their algorithms can locate these obstacles by observing expert demonstrations.

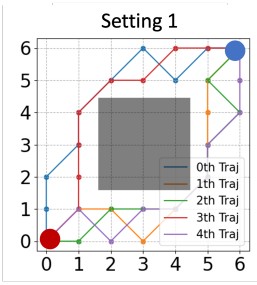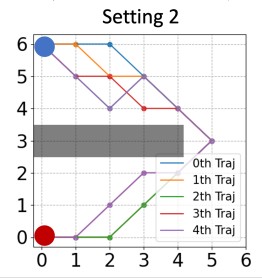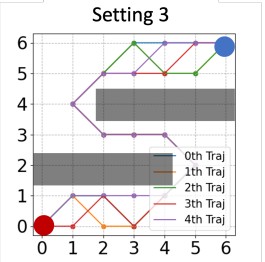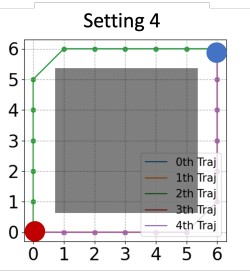

Figure 7: The Grid-World Environments: We randomly select five expert trajectories (denoted as $0^{th}$ Traj to $4^{th}$ Traj) for visualization purposes. The starting point and destination are denoted by red and blue circles, respectively. The primary objective of the ICRL algorithms is to deduce the constrained region, which is unobservable and illustrated by the gray color.

Figure 7 presents four distinct Grid-World environments. These environments have been chosen for their ease of visualization and result analysis, and it is relatively convenient to expand these simple gird-worlds into more complicated scenarios, for example, Baert et al. (2023); Xu & Liu (2024b) developed Grid-World environments with stochastic transition dynamics, while Qiao et al. (2023) added multiple types of constraints to Grid-Worlds. While Grid-World environments, with their low-dimensional and discrete state spaces (represented by x-y coordinates), offer several key benefits, they present a challenge in generalizing model performance to the environment with high-dimensional and continuous states. By explicitly considering the stochasticity, MCEnt-ICRL algorithms (Section 5) outperform those based on MEnt framework (Section 4).

**Virtual Environments.** Evaluating RL algorithms directly in real applications can be inefficient, costly, and potentially lead to critical safety issues. As an alternative, simulated virtual environments offer an

effective platform for testing RL algorithm performance. These environments enable episodic replay and efficient exploration, mitigating many of the challenges associated with real-world applications.

Among various game simulators, MuJoCo (Todorov et al., 2012) has been extensively employed to evaluate the performance of ICRL in terms of recovering location constraints in robot control tasks (Malik et al., 2021; Liu et al., 2023; Qiao et al., 2023). For instance, if an agent observes that certain locations are never visited by expert agents, it can reasonably infer that these locations are likely to be unsafe. To construct proper virtual environments for validating ICRL algorithms, Liu et al. (2023) modified some popular MuJoCo environments (including Half-cheetah, Ant, Inverted Pendulum, Walker, Hopper and Swimmer, see Figure 8) by incorporating some predefined constraints into each environment. Table 3 summarizes the environment settings. The constraints are added to the $X$-coordinate, moving velocity, and angular velocity of the robot body under different controlling tasks. During the evaluation, instead of directly observing these added constraints, the agent has access to the expert demonstrations and the goal is to recover these constraints. Within these environments, MEnt-ICRL (Malik et al., 2021) was the first to propose inferring constraints represented by a neural network to accommodate continuous features. Subsequent studies, including VICRL (Liu et al., 2023), UA-ICRL (Xu & Liu, 2024a), and AR-ICRL (Xu & Liu, 2024b), have explored stochastic dynamics by introducing noise into the transition functions of the environments. Specifically, in environments such as Blocked Half-Cheetah, Blocked Ant, and Crippled Walker, Xu & Liu (2024b) examined various types of noise, including fully random noise, partially random noise, and adversarial noise. Utilizing a robust optimization framework, AR-ICRL has demonstrated superior performance under these diverse noise conditions.

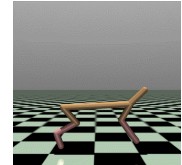 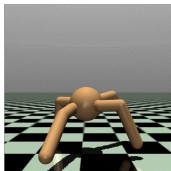 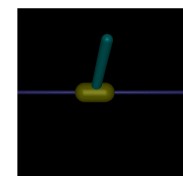 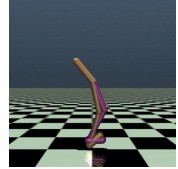 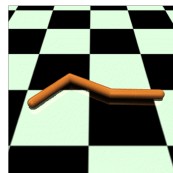

Figure 8: Mujoco environments. From left to right, the environments are Half-cheetah, Ant, Inverted Pendulum, Walker, and Swimmer.

Table 3: The virtual environments with different types of constraints.

| Name | Constraint Types | Obs. Dim. | Act. Dim. | Constraints |
|---|---|---|---|---|
| Position-Blocked Half-cheetah | Spatial Constraint | 18 | 6 | X-Coordinate $\geq$ -3 |
| Position-Blocked Ant | Spatial Constraint | 113 | 8 | X-Coordinate $\geq$ -3 |
| Leg-Blocked Ant | Kinematic Constraint | 113 | 8 | Leg Angular Velocity $\leq$ 1 |
| Limited-Speed Ant | Dynamic Constraint | 113 | 8 | Moving Speed $\leq$ 0.5 |
| Position-Biased Pendulum | Spatial Constraint | 4 | 1 | X-Coordinate $\geq$ -0.015 |
| Position-Blocked Walker | Spatial Constraint | 18 | 6 | X-Coordinate $\geq$ -3 |
| Crippled Walker | Kinematic Constraint | 18 | 6 | $\|$Thigh Angle$\| \leq$ 0.6 |
| Limited-Speed Walker | Dynamic Constraint | 18 | 6 | Moving Speed $\leq$ 1 |
| Position-Blocked Swimmer | Spatial Constraint | 10 | 2 | X-Coordinate $\leq$ 0.5 |
| Leg-Blocked Hopper | Kinematic Constraint | 18 | 6 | Leg Angular Velocity $\leq$ 0.3 |

**Realistic Environment.** Realistic environments denote RL environments whose dynamics are grounded by real-world datasets. Under this setting, Liu et al. (2023) formulated a Highway Driving (HighD) environment (Figure 9). The key objective of the HighD environment was to investigate whether an agent is capable of inferring the constraints adhered to by human drivers and safely navigating a self-driving car, referred to as the 'ego car', to its destination.

Specifically, Liu et al. (2023) utilized a HighD dataset (Krajewski et al., 2018) that records naturalistic vehicle trajectories from German highways. Within each scenario, HighD contains information about the static background (e.g., the shape and the length of highways), the vehicles, and their trajectories. The HighD environment is constructed by randomly selecting a scenario from the dataset and an ego car for control in this scenario. The game context, which is constructed by following the background and the trajectories of other vehicles, reflects the driving environment in real life. The observed features are collected

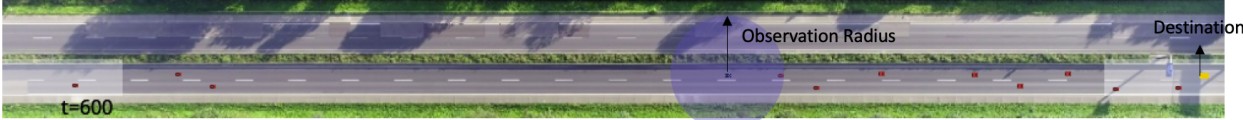

Figure 9: (Figure 5 in (Liu et al., 2023)) The Highway Driving (HighD) environment. The ego car is blue, other cars are red. The ego car can only observe the things within the region (marked by blue). The goal is to drive the ego car to the destination (in yellow) without going off-road, colliding with other cars, or violating time limits and other constraints (e.g., speed and distance to other vehicles).

by the CommonRoad-RL toolkit (Wang et al., 2021). Table 4 summarized the studied constraints, including a car speed constraint and a car distance constraint which ensures the ego car can drive at a safe speed and keep a proper distance from other vehicles.

Table 4: The constraints for realistic environments (Table 3 in (Liu et al., 2023)).

| Type | Name | Constraint Types | Obs. Dim. | Act. Dim. | Constraints |
|---|---|---|---|---|---|
| Realistic | HighD Velocity Constraint | Dynamic Constraint | 76 | 2 | Car Velocity $\leq$ 40 m/s |
| | HighD Distance Constraint | Dynamic Constraint | 76 | 2 | Car Distance $\geq$ 20 m |

## 9.2 Applications

In this section, we introduce the potential applications of ICRL in solving practical problems.

**Autonomous Driving.** Designing autonomous agents to control vehicles on open roads is a challenging task, particularly when considering the long-tail safety-critical events (Yan et al., 2023). Learning a policy that can develop a secure driving strategy across diverse driving scenarios is difficult without explicitly modeling safety constraints. However, the optimal constraints for autonomous driving should be context-sensitive and attuned to driving behaviors, aspects often unknown in real-world applications. On the other hand, there has been a significant release of high-quality, open-source datasets that document naturalistic vehicle trajectories in various key driving scenarios. These include highways (Krajewski et al., 2018), intersections (Bock et al., 2020; Zhan et al., 2019), roundabouts (Krajewski et al., 2020), and entrances and exits of highways (Moers et al., 2022). These recorded trajectories serve as a testament to the expertise of human driving behaviors. As such, the application of ICRL to infer constraints from human demonstrations becomes an essential aspect of this field.

**Embodied AI.** Embodied AI refers to artificial intelligence systems that interact with the physical world through sensors and actuators, enabling them to perform tasks in real-world environments. As a crucial component of embodied AI, generalizable policy learning represents a significant area of interest within the robotics and artificial intelligence communities. Many real-world robotic applications require safety guarantees during the design of control policies, which not only optimize task performance but also adhere to some implicit constraints imposed by safety, ethics, or operational requirements, such as safe navigation, human-robot interaction, and robot manipulation (Dulac-Arnold et al., 2021; Brunke et al., 2022; Zhao et al., 2023). While tasks in robotics typically have specific objectives, such as reaching a destination or handling an item, their underlying constraints often remain ambiguous. For instance, a robot may need to maintain specific poses (i.e., natural motions (Hansen et al., 2024)) during operation or keep certain distances from people and objects to avoid areas that are not explicitly defined. In these cases, algorithms must infer these unknown constraints by observing expert demonstrations and exploring the environment with a known reward function. An immediate approach to accomplishing this task is ICRL, which infers these constraints and learns a policy that adheres to them, thereby enabling robots to align their actions with desired behaviors, ensuring safer and more efficient interactions across different environments. For example, Palafox et al. (2023) proposed a game-theoretic approach to multi-robot collision avoidance, utilizing rotating hyperplane constraints that are learned from expert demonstrations.

**Decision Making in Healthcare.** Recent advancements in AI healthcare have explored the application of RL to develop AI assistants for diagnosis and disease treatment (Yu et al., 2021). However, the policies derived from these applications can sometimes lead to unsafe behaviors, such as administering excessive drug dosages, making inappropriate adjustments to medical parameters, or implementing abrupt changes in medication dosages. To learn safe decisions, a common method for learning safe policies is CRL, but the success of CRL significantly relies on the accurate and trustworthy representation of constraints. However, in healthcare, designing the constraints based solely on prior knowledge is challenging. The effectiveness of many healthcare applications depends on integrating the underlying experience of medical experts. In this context, ICRL provides a promising method for extracting constraints from expert demonstration data thereby inducing more reliable decision-making systems in Healthcare.

**Sports Analytics.** In professional team sports, winning the game often involves complex interactions between multiple players who execute a series of movements within a confined play court and limited playtime. This scenario shares numerous similarities with the properties of MDPs, making RL a compatible method for modeling the behavior of professional athletes. Several previous studies (Decroos et al., 2019; Liu & Schulte, 2018; Liu et al., 2020) have leveraged policy evaluation algorithms to assess the individual contributions of each player towards maximizing specific rewards, such as the probability of winning the game. However, in real-world situations, players often prioritize their short-term preferences over the specified long-term goals. Accordingly, a significant challenge in sports analytics is to gain an understanding of and provide explanations for the behavior of these professional athletes, including the motivations behind their movements (Albert et al., 2017). A previous study (Luo et al., 2020) expanded on MEnt IRL to infer the rewards function. With this foundation, ICRL algorithms can be applied to infer the constraints that professional players adhere to. For precise inference of constraints, it's crucial for the algorithm to account for both the cooperative behaviors of teammates and the competitive actions of opponents. As such, sports analytics can serve as a significant application for assessing the efficacy of multi-agent ICRL algorithms.

## 10 Conclusion

As an important research topic for enabling safe control, ICRL has received considerable attention in recent years. This paper systematically defines the problem of ICRL while distinguishing it from related research topics and summarizes the recent progress in conducting ICRL under discrete and continuous environments, from a limited demonstration dataset and multiple expert agents. We introduce the benchmark and potential applications of ICRL. To facilitate future research, we outline some open questions regarding ICRL:

### 10.1 Open Questions in Short-Term

While these open questions remain unexplored, we can still refer to a variety of established techniques, methods, and methodological frameworks for guidance. These encompass the following subjects:

**Game-Theoretic Multi-Agent Constraint Inference.** In the process of inferring constraints from a multi-agent environment, prior methodologies (Liu & Zhu, 2022) typically assume a cooperative scenario where each agent adheres to unique individual constraints. Furthermore, the behavior satisfying one agent's constraints is presumed not to conflict with the actions of others. However, in real-world applications, such as autonomous driving in open traffic, drivers often engage in competitive interactions, particularly under conditions of heavy traffic (Ding et al., 2023). The movement of one vehicle can significantly affect the likelihood of other vehicles reaching their distance constraints. In such circumstances, the task of constructing a game-theoretical model to capture the competitive behavior of agents and determining the potential equilibrium among the policies (i.e., strategies) of different agents is an important direction of future work. A potential solution involves extending existing RL solvers from general-sum games (Bai et al., 2020; Song et al., 2022) to ICRL scenarios. However, these methods primarily focus on the forward control problem. Incorporating them into inverse constraint inference remains a substantial challenge.

**Inverse Constrained Learning from Offline Dataset.** Traditional ICRL algorithms typically adopt an online learning paradigm, wherein the agent iteratively amasses experience through interaction with the environment and leverages that experience to infer constraints and update its policy. However, in numerous

scenarios, online interaction may be impractical, primarily due to the high cost of data collection (for instance, in robotics, educational agents, or healthcare), or potential risks (such as in autonomous driving or healthcare) (Levine et al., 2020). Conversely, many real-world datasets are already replete with rich experiences from both expert and sub-optimal agents. Consequently, a more pragmatic approach in ICRL involves learning constraints from offline datasets, independent of environmental interactions. In this context, Quan et al. (2024) introduced an offline ICRL algorithm for learning hard constraints while the problem of inferring soft constraints remains unresolved. Besides, empirical results indicate that performance is unstable and highly sensitive to the choice of hyperparameters and the underlying distance metric (e.g., $\mathcal{D}_f$ in Equation 55). Additionally, the composition of offline datasets—including the proportion of expert, constraint-violating, and constraint-satisfying but sub-optimal controlling trajectories—significantly impacts model performance. Developing a robust method to reliably infer constraints continues to pose a significant challenge in offline ICRL. Potential solutions include leveraging more stable and robust methods from offline inverse RL (Fu et al., 2017; Zeng et al., 2023) for learning constraints.

**Inferring Generalizable Constraints.** In real-world scenarios, the dynamics of deploying environments can be disrupted by unforeseen noise or unpredictable uncertainties inherent to the system. Without considering these disturbances, the ICRL algorithm's reliability could be significantly compromised, particularly in safety-critical applications such as autonomous driving or robotic surgery. These disturbances can greatly vary across different systems. For instance, in the task of highway merging (refer to Figure 1), systems from different locations may be influenced by diverse disturbances stemming from weather conditions, temperature variations, and local driving styles. To handle these challenges, Xu & Liu (2024b) devised an ICRL algorithm under the robust optimization framework (Ben-Tal et al., 2009). However, this model primarily focuses on the mismatched transition functions between the training and deployment environments. It does not account for other critical factors, such as mismatched rewards and the resulting changes in occupancy measures, which are also crucial for comprehensive robustness assessment. To solve these questions, it is significant to generalize the existing robust RL (Wang & Zou, 2021; Panaganti et al., 2022; Wang et al., 2024a) and inverse RL (Viano et al., 2021; Wei et al., 2023) frameworks to the ICRL setting. More importantly, the generalization of these learned constraints to real-world control scenarios has not been extensively studied. The test beds are mostly based on simulated environments rather than real-world settings. A crucial direction for future research is to demonstrate the effectiveness of these methods in real-world control applications, such as facilitating collision-free robot manipulation policies.

**Inferring Dynamic Constraints.** Previous ICRL studies typically focus on learning static constraints from offline expert demonstrations. However, in the real world, such static constraints may be ineffective because of the dynamic and ever-changing nature of environments. Constraints in practical scenarios often depend on time-sensitive factors such as moving obstacles, interactions with other agents, or shifts in environmental conditions (e.g., weather or lighting changes). Static constraints fail to capture this variability, which can lead to suboptimal or unsafe behavior when deploying learned policies in dynamic settings. For instance, in autonomous driving, the safe distance between vehicles depends on their relative velocities, traffic density, and weather conditions. A static constraint inferred from the offline dataset might indicate a fixed minimum distance, which could be either overly conservative or dangerously lax under specific conditions. Similarly, in robotic manipulation, obstacles may move unpredictably, and the robot must adjust its constraints dynamically to avoid collisions while completing a task efficiently. Dynamic constraints require more complicated modeling approaches that account for temporal changes and adapt in real-time. Unlike static constraints, they must reflect the evolving feasibility of actions based on the current state, past trajectory, and expected future conditions. Future work could leverage sequence modeling approaches, such as recurrent neural networks (RNNs) (Medsker et al., 2001) and Transformers (Vaswani et al., 2017) to capture the temporal evolution of constraints. Furthermore, employing online inverse RL (Self et al., 2020; Lian et al., 2021; Wang & Zou, 2021) techniques to design online ICRL algorithms capable of iteratively refining constraints based on new observations also serves as a promising research direction.

**Constraint Inference for LLMs.** The concept of constraint inference, traditionally rooted in RL, has significant potential in aligning large language models (LLMs) with specific ethical, safety, or operational requirements. LLMs, such as GPT-style models (Floridi & Chiriatti, 2020; Achiam et al., 2023), generate outputs based on learned patterns in vast corpora of text. However, this capability comes with challenges

in ensuring that their responses adhere to desired constraints, especially in safety-critical applications like healthcare and content moderation (Wei et al., 2024a; Kumar et al., 2023; Liu et al., 2024). Enforcing proper constraints is a crucial strategy for preventing LLMs from generating harmful content. Recent studies, such as Dai et al. (2024), have explored the use of Lagrange methods to align LLMs with safety requirements. However, instead of relying on a cost model learned from preferences and subsequently used for performing forward-constrained optimization, an intriguing alternative is to infer these constraints directly from observed data without preferences. This approach establishes a meaningful connection between reinforcement learning from human feedback (RLHF) and ICRL. By leveraging constraint inference techniques, LLMs can be more effectively aligned with safety and ethical guidelines, ensuring their outputs remain within well-defined and adaptable boundaries.

### 10.2 Open Questions in Long-Term

Although the methods for realizing these long-term objectives are not fully explored, these issues are critical for the future development of ICRL.

**Theoretical Grounding for ICRL.** Many recent works have developed theoretical understandings of IRL. Notably, to handle the identifiability issue, Kim et al. (2021) studied how identifiability relates to properties of the MDP model and proved necessary and sufficient conditions for identifiability in deterministic MDP with the MEnt-RL objective. Alternatively, Lindner et al. (2022) defined the feasible reward set featuring the region of rewards that can equivalently explain the expert behaviors. A continuing work by Metelli et al. (2023) formally introduced the problem of estimating the feasible reward set, the corresponding PAC requirement, and the minimax lower bound on the sample complexity.

In the ICRL problems, it is crucial to understand the identifiability of learned constraints and the convergence of inference. Yet, existing research primarily focuses on inferring rewards rather than constraints (Metelli et al., 2021; Lindner et al., 2022; Metelli et al., 2023). Contrary to the unconstrained IRL problems, the optimality of ICRL is interconnected with the nature of constraints (including hard, soft, or probabilistic constraints) and the chosen constrained optimization method, such as the Lagrange method or Interior Point Methods. Defining the precise identifiability conditions and the feasible constraint set can be quite challenging. Recently, an ICRL study (Yue et al., 2024) defined the feasible constraint set for ICRL problems. However, their focus was primarily on simplified discrete environments. Therefore, how to develop novel strategies tailored to the unique intricacies of ICRL still remains a major challenge.

**Learning Physic-Realistic Constraints.** Recent advancements in generative diffusion models, such as OpenAI's Sora[4] and Google's Veo[5], have shown remarkable performance in video generation. However, critical challenges remain in applying these token-by-token sequential generators to simulate real-world scenarios. These data-driven simulators often produce videos that violate fundamental physical laws and common rules observed in the real world. Recent advancements in embodied AI simulators have highlighted a critical issue: without ensuring physical realism, skills learned in simulated environments may not be transferable to realistic settings. This gap underscores the importance of developing simulations that accurately mimic real-world physics to effectively guide robots in practical applications (Hua et al., 2024; Nasiriany et al., 2024; Wang et al., 2024b). To address these challenges, a key direction for future work at ICRL involves extracting these laws and rules from real-world videos by treating them as expert demonstrations and representing the physical rules through constraints. Consequently, developing a suitable representation for these constraints and a reliable approach to inferring them from real-world videos presents significant challenges.

### Acknowledgments

This work is supported in part by Shenzhen Fundamental Research Program (General Program) under grant JCYJ20230807114202005, Guangdong-Shenzhen Joint Research Fund under grant 2023A1515110617, Guangdong Basic and Applied Basic Research Foundation under grant 2024A1515012103, and Guangdong Provincial Key Laboratory of Mathematical Foundations for Artificial Intelligence (2023B1212010001).

---

[4] https://openai.com/index/sora/
[5] https://deepmind.google/technologies/veo/

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
