# OpenReview forum: "A Comprehensive Survey on Inverse Constrained Reinforcement Learning: Definitions, Progress and Challenges"
_TMLR — Accepted by TMLR_

### Review · Reviewer_4Ecj · 2024-10-18

**Summary Of Contributions:**

This paper presents to the best of my knowledge the first survey in the area of constraint inference. The authors discuss recent developments in constraint inference in reinforcement leaning but also in control tasks.The authors provide motivation for the research area and they follow to discuss most of the recent advances. They also discuss some open problems as a possible direction for future research.

**Audience:**

Yes

**Claims And Evidence:**

Yes

**Requested Changes:**

It would be great if the authors could address the weaknesses listed above.  Regarding the third bullet point, there is no need for trying to derive computational complexities, but some discussion or table with some comments would be a good addition to the paper.

**Strengths And Weaknesses:**

**Strengths**:

The paper is well structured. The authors try to present some key details about the setup and implementation of each paper in an organized way. To the best of my knowledge most of the relevant research work has been included. I have included a few additional papers later in my review that the authors might want to consider. The authors also try to shed some light on the possible future directions that might be of interest.

**Weaknesses**:
* Surveys that also benchmark different methods tend to be even more valuable in my view, although I understand that this is a significantly more involved task. I think that regarding the future direction the authors should try and elaborate a bit more on the "challenges part" and possible future research avenues. For example, the case of inferring dynamic constraints (like moving obstacles) could be discussed. Also for example, constraint inference could be potentially utilized in aligning LLMs so that they respect constraints.

* The authors also do not mention anything about the complexity of the methods. In many cases such arguments were not done in the original papers and so it is hard to tabulate complexities for different methods, but I think that some comments can be added regarding that.

Some additional references that you might want to consider are listed at the end.

In the section "Inverse Constrained Learning from Offline Dataset" I believe that except for Qual et al. constraint inference from offline data has been studied in [1].

Here are some typos/rewordings/minor fixes for your convenience.

In (32) and (41) the integration is over $a$ but a does not appear in the integral.

In (50) three different datasets appear. What is the relationship of $\mathcal{D}$ with the others?

"By utilizing the posterior distribution over constraint sets at previous..." I think this sentence can be rewritten to be a bit more clear

In (52) the subscripts in the expectations appear to be in bold for some reason

Also it would be nice to start a section with text and not with Figures. This happens in section1 and 2.4.

In (5) maybe include $\forall i \in [1,N]$ as in (6)

In (11) it is not clear which is the original problem of this dual. I assume it is (5)

"task of inferring the implicit constraints followed by expert agents" maybe a different word than followed is more appropriate

Sometimes when referring to a single person "and inherent to experts’ own experience" it should be expert's. This happens a few times in the paper.

"Discounted factor" should be Discount factor

The term nominal is used a few times in the paper but it is not clearly defined somewhere.

where $\omega_m\sim q(\omega)$ Specifically, there is a . missing at the end of the sentence.

"constraints function $c(s,a). i.e. \pi^{\epsilon}\mathbb{E} =$ ..." avoid starting a sentence with i.e.

"Within each partition ι, CBN-IRL learns" in multiple sections in the paper the letter i appears without the dot on top. I think it would be clearer if you used just i. If it is done because you want to use i only for the index of a constraint then it is fine.

"proposed to learn single reward function" --> to learn a single

"here f is implemented by a multi-layer" $f$ or $f_{\omega}$?

I would avoid having equations between text, especially when they affect the spacing between lines. In those case just use write them as an equation. See for example below (56). Also when writing an optimization problem use a different line for the constraint (the s.t. part)

"., locating outside the support of" -->located

In (59) there is both $\pi$ and $\pi(\theta)$ appearing. Is this a typo?

In (61) and a few other places there is no gap between the $argmax$ and the expectation

"for extracting expert constraints from their demonstration data" maybe rephrase to constraint from expert demonstration data?

" independent of environmental interaction." --> interactions

sometimes a colon is used to introduce an equation bit sometimes it is missing. (for example (53) and (54)). Also there is no need for a colon to introduce definition 6.1

In (18) maybe just use $|C|$ for the size of the set

 "Striving for reliable constraint inference, the policy represent" maybe you meant representation?

 "domains, 2) soft constraints" instead of , maybe use and

 "In the continuous environment, constructing"--> environments

In definition 5.1 is should be $s_T$

"but the soft constraints requires a threshold"--> require

"by expert demonstration is a trivial "--> demonstrations

"Lagrange method or Interior Points Method."--> Interior Point Methods

 "The parameters of the constraint models $\omega$ is given by:"--> are

 in page 4 when the cost $c$ is discussed I think it would be clearer if it was written as $c(s,a)$ as it is a function. That appleid for example in table 2 and above eq. (2)

In (19) the partition should be $Z_c$ or $Z_{c_{\omega}}$?

"in RL to learn model-free controller without"--> a model-free...

" While studying these topics are not our primary focus, we"--> is not



**References**:

[1] D. Papadimitriou et al. "Bayesian Constraint Inference from User Demonstrations Based on Margin-Respecting Preference Models", ICRA 2024

[2] C. Basich et al. "Learning Constraints on Autonomous Behavior from Proactive Feedback", IROS 2023

[3]. J. Jang et al. "Inverse Constraint Learning and Generalization by Transferable Reward Decomposition", RA-L

[4] K. Kim et al. "Learning Shared Safety Constraints from Multi-task Demonstrations" NIPS 2023

[5] F. Palafox et al. "Learning Hyperplanes for Multi-Agent Collision Avoidance in Space"

[6] Q. Wei et al. "Isoperimetric Constraint Inference for Discrete-Time Nonlinear Systems Based on Inverse Optimal Control", TCYB

[7]  F. Ding et al, "X-MEN: Guaranteed XOR-Maximum Entropy Constrained Inverse Reinforcement Learning", UAI 2022

[8] B. Peng et al. "Positive-Unlabeled Constraint Learning (PUCL) for Inferring Nonlinear Continuous Constraints Functions from Expert Demonstrations", arxiv

[9]  B. Yue et al. "Provably Efficient Exploration in Inverse Constrained Reinforcement Learning", arxiv

---

> ### Author Response · Authors · 2024-11-27
> **Author Response to Reviewer 4Ecj - Part 1**
>
> Dear Reviewer, we sincerely thank you for dedicating your time and effort to review our paper. Your insightful comments and constructive feedback have been invaluable in improving the quality and clarity of our work. We hope our responses below can address your concerns effectively.
>
> > *Comment 1. Surveys that also benchmark different methods tend to be even more valuable in my view, although I understand that this is a significantly more involved task.*
>
> **Response 1.** Thanks for pointing this out. In this work, our primary objective is to provide a comprehensive survey that outlines the definitions, progress, and challenges in ICRL. Given the breadth of recent developments in this area, we focus on systematically organizing and synthesizing advancements to assist researchers in navigating the growing literature. However, we recognize that adding benchmarks is a valuable extension. Notably, an ICRL benchmark has already been introduced, as described in Section 9.1, which proposed some meaningful tasks for validating ICRL algorithms. However, since different ICRL algorithms are designed with varying settings and objectives, **it is challenging to compare them fairly within a unified framework.** In future work, we aim to enhance this benchmark by incorporating a broader range of settings and facilitating the implementation and evaluation of more advanced ICRL algorithms.
>
> ---
>
> > *Comment 2. I think that regarding the future direction the authors should try and elaborate a bit more on the "challenges part" and possible future research avenues. For example, the case of inferring dynamic constraints (like moving obstacles) could be discussed. Also for example, constraint inference could be potentially utilized in aligning LLMs so that they respect constraints.*
>
> **Response 2.** Thank you for your valuable suggestions. We have expanded the discussion on the challenges and proposed more potential future research directions, including dynamic constraints and applications in the context of LLMs. Please refer to **Section 10** for detailed updates, which are **highlighted in blue**.
>
> ---
>
> > *Comment 3. The authors also do not mention anything about the complexity of the methods. In many cases such arguments were not done in the original papers and so it is hard to tabulate complexities for different methods, but I think that some comments can be added regarding that. Regarding the point, there is no need for trying to derive computational complexities, but some discussion or table with some comments would be a good addition to the paper.*
>
> **Response 3.** Thank you for this valuable suggestion. We agree that providing a discussion of complexity can help readers better compare the various ICRL algorithms. We have spent effort to create a table listing the computational complexities of different methods. However, this endeavor faced challenges, primarily because **ICRL algorithms are based on differing settings and target different goals.**
>
> For example, in the discrete state-action space, ICRL algorithms often scan all states to locate the constraint set. However, this approach becomes intractable in continuous settings. Additionally, variations such as multi-agent ICRL and simultaneous reward and constraint inference have different objectives, which further complicates direct comparisons.
>
> Given these complexities, we have determined that a direct comparison of computational complexities may not always be informative or helpful for the reader. We have also considered discussing the complexity within the context of each specific setting. However, as we typically introduce only one algorithm per setting, this approach would not effectively convey the computational demands across different scenarios.
>
> ---
>
> > *Comment 4. Some additional references that you might want to consider are listed at the end.*
>
> **Response 4.** Thanks for pointing out these related works. We have included and discussed them accordingly in the revised manuscript. Specifically, the references you mentioned (i.e., Ref. [1-9]) are **discussed in Sections 6.3, 2.3, 7.3, 8.1, 9.2, 3.2, 3.1, 4.2, and 10.2**, respectively. The modifications are **highlighted in blue**.
>
> ---
>
> > *Comment 5. In the section "Inverse Constrained Learning from Offline Dataset" I believe that except for Quan et al. constraint inference from offline data has been studied in [1].*
>
> **Response 5.** Thanks for pointing this out. We have included the mentioned paper in this section accordingly (Section 6.3).

---

> ### Author Response · Authors · 2024-11-27
> **Author Response to Reviewer 4Ecj - Part 2**
>
> > *Comment 6. Here are some typos/rewordings/minor fixes for your convenience.*
>
> **Response 6.** Thank you very much for your thorough review of our paper. We have thoroughly reviewed all the potential errors you identified and have updated the paper in line with your suggestions. Each modification has been addressed individually and is highlighted **in blue** in the revised paper for clarity:
>
> | No. | Comment | Response |
> |---|---|---|
> | 1 | In (32) and (41) the integration is over $a$ but $a$ does not appear in the integral. | We have corrected them; please check (33) and (42). |
> | 2 | In (50) three different datasets appear. What is the relationship of $\mathcal{D}$ with the others? | We have corrected the notation; please check the words under Eq (51). |
> | 3 | "By utilizing the posterior distribution over constraint sets at previous..." I think this sentence can be rewritten to be a bit more clear | We have modified it to be clearer; please check the words above Eq (43). |
> | 4 | In (52) the subscripts in the expectations appear to be in bold for some reason. | We have corrected it; now in (53). |
> | 5 | Also it would be nice to start a section with text and not with Figures. This happens in section1 and 2.4. | We have corrected them to make the text first, accordingly. |
> | 6 | In (5) maybe include $\forall i\in[1,I]$  as in (6). | We have corrected it accordingly. |
> | 7 | In (11) it is not clear which is the original problem of this dual. I assume it is (5). | We have modified it accordingly for consistency. |
> | 8 | "task of inferring the implicit constraints followed by expert agents" maybe a different word than followed is more appropriate | We have rewritten the sentence; please check the first sentence of the abstract. |
> | 9 | Sometimes when referring to a single person "and inherent to experts’ own experience" it should be expert's. This happens a few times in the paper. | We have corrected experts' to expert's accordingly in the paper. |
> | 10 | "Discounted factor" should be Discount factor. | We have corrected it; please check Table 2. |
> | 11 | The term nominal is used a few times in the paper but it is not clearly defined somewhere. | We have clarified it, please check Table 2. |
> | 12 | where $\omega_m\sim q(\omega)$ Specifically, there is a . missing at the end of the sentence. | We have added a full stop there. |
> | 13 | "constraints function $c(s,a). i.e.$..." avoid starting a sentence with i.e. | We have modified it; please check the words above (54). |
> | 14 | "Within each partition $\iota$, CBN-IRL learns" in multiple sections in the paper the letter i appears without the dot on top. I think it would be clearer if you used just i. If it is done because you want to use i only for the index of a constraint then it is fine. | Thanks for pointing out this. Indeed $\iota$ denotes partition of trajectories in the context, so we may not use the index $i$ for representing it. |
> | 15 | "proposed to learn single reward function" should be to learn a single. | We have corrected it accordingly. |
> | 16 | "here $f$ is implemented by a multi-layer" $f$ or $f_\omega$? | It should be $f_\omega$, we have clarified it. |
> | 17 | I would avoid having equations between text, especially when they affect the spacing between lines. In those case just use write them as an equation. See for example below (56). Also when writing an optimization problem use a different line for the constraint (the s.t. part). | Thanks for pointing this out. We have revised the paper to avoid such cases. |
> | 18 | "., locating outside the support of" --> located. | We have corrected it. |
> | 19 | In (59) there is both $\pi$ and $\pi_\theta$ appearing. Is this a typo? | It should all be $\pi$. We have fixed it; now in (66). |
> | 20 | In (61) and a few other places there is no gap between the $\arg\max$ and the expectation. | We have tried to leave some space, which is likely due to the compilation format. |
> | 21 | "for extracting expert constraints from their demonstration data" maybe rephrase to constraint from expert demonstration data? | Yes, we have corrected it. |
> | 22 | "independent of environmental interaction." --> interactions. | We have corrected it. |
> | 23 | sometimes a colon is used to introduce an equation bit sometimes it is missing. (for example (53) and (54)). Also there is no need for a colon to introduce definition 6.1. | We have unified the paper to use the colon for introducing an equation. And we have changed the colon to a full stop before definition 6.1. |
> | 24 | In (18) maybe just use $\|C\|$ for the size of the set. | We have corrected it; now in (19). |
> | 25 | "Striving for reliable constraint inference, the policy represent" maybe you meant representation? | Yes, we have corrected it. |
> | 26 | "domains, 2) soft constraints" instead of , maybe use and | We have added an "and". |

---

> ### Author Response · Authors · 2024-11-27
> **Author Response to Reviewer 4Ecj - Part 3**
>
> | No. | Comment | Response |
> |---|---|---|
> | 27 | "In the continuous environment, constructing" --> environments. | We have corrected it. |
> | 28 | In definition 5.1 is should be $s_T$. | We have corrected it. |
> | 29 | "but the soft constraints requires a threshold" --> require. | We have corrected it. |
> | 30 | "by expert demonstration is a trivial" --> demonstrations. | We have corrected it. |
> | 31 | "Lagrange method or Interior Points Method." --> Interior Point Methods. | We have corrected it. |
> | 32 | "The parameters of the constraint models $\omega$ is given by:" --> are. | We have corrected it. |
> | 33 | in page 4 when the cost $c$ is discussed I think it would be clearer if it was written as as it is a function. That appleid for example in table 2 and above eq. (2). | We have corrected it for alignment with Table 2. Please check the first line of Page 5. |
> | 34 | In (19) the partition should be $Z_c$ or $Z_{c_\omega}$? | It should be $Z_{c_\omega}$, we have corrected it; now in (20). |
> | 35 | "in RL to learn model-free controller without" --> a model-free... | We have corrected it. |
> | 36 | "While studying these topics are not our primary focus, we" --> is not. | We have corrected it. |
>
> ---
> We hope the above response could resolve your concerns. If you have more questions, please feel free to discuss them with us. Thank you once again for your invaluable review and the thoughtful effort you invested in reviewing our paper!

---

> ### Author Response · Authors · 2024-12-10
> **Kind reminder regarding your feedback on TMLR paper 3369**
>
> Dear Reviewer 4Ecj,
>
> We hope this message finds you well. We understand this might be a particularly busy time for you, and we genuinely appreciate the time you dedicate to reviewing.
>
> We would like to kindly remind you to review our rebuttal at your convenience. Your valuable feedback is instrumental in helping us refine and improve our paper. Should there be any specific points that require further clarification, please do not hesitate to let us know.
>
> Thank you for your time, effort, and thoughtful insights. We sincerely appreciate your contributions to this process.
>
> Best regards,
>
> Authors of TMLR Paper 3369

---

> ### Comment · Reviewer_4Ecj · 2024-12-15
> **Thank you for addressing my comments**
>
> Thank you for the time you spent addressing my comments and suggestions. I think that the paper has improved significantly after incorporating the feedback from all of the reviewers. I think that the additional figures and comments have made the paper easier to follow.
>
> I found a few more typos that I list here for your convenience:
>
> are subject to optimize at-->to optimization at
>
> structure of ICRL methdos-->methods
>
> we introduce the ICRL methods that models --> method

---

> ### Author Response · Authors · 2024-12-15
> **Thanks for your feedback!**
>
> Thank you very much for the time and effort you invested in reviewing our manuscript. We have revised our manuscript to fix the typos. Your detailed and insightful comments have been extremely valuable in guiding us toward significant improvements in both the quality and clarity of our paper!

---

### Review · Reviewer_PDYr · 2024-11-01

**Summary Of Contributions:**

This paper offers a comprehensive survey of recent developments in Inverse Constrained Reinforcement Learning (ICRL), focusing on how implicit constraints can be inferred from expert demonstrations. The authors begin by establishing foundational definitions and draw comparisons between ICRL and related fields like Inverse Reinforcement Learning (IRL) and Inverse Optimal Control (IOC). They systematically categorize ICRL methods across various settings, including deterministic and stochastic environments, multi-agent scenarios, and cases with limited or partial demonstrations.

**Audience:**

Yes

**Broader Impact Concerns:**

No concerns about broader impacts.

**Claims And Evidence:**

Yes

**Requested Changes:**

1. At the end of Section 2.4, could you clarify whether "Interpretable Constraints" refers to a regularization method? Is φ(s, a) a learned model or a fixed function? How does learning $\phi(s, a)$ differ from learning $c(s, a)$, and how does this relate to interpretability?
2. In Section 3.2, you mention that "ICRL can incorporate recent advancements in RL to learn model-free controllers without knowing or estimating the model dynamics (i.e., model-free methods)." Does this imply that ICRL focuses only on model-free methods, or does it also consider model-based methods?
3. Please clarify the terminology in Section 4.2. Is $\phi_w(s, a)$ a permissibility function? Earlier, a permissibility function is defined as "a function learned to distinguish between feasible and infeasible states within these regions." Does the concept of a permissibility function apply only to states, or can it also apply to state-action pairs?
4. In Section 6, the issue of epistemic uncertainty seems closely related to the false correlation problem in offline RL. It would be beneficial to add a brief discussion and reference to the false correlation problem.
5. In Section 8, you mention three levels of Multi-Agent ICRL. Could you elaborate on the key differences between the third level and the first two? Do the first two levels consider agents operating sequentially rather than simultaneously?
6. In Section 10, you discuss open questions in ICRL. It would be helpful to cite some papers that have attempted to address these questions, even if they aren't specific to ICRL, since these challenges are common in other RL domains.

**Strengths And Weaknesses:**

**Strengths**

- The paper covers a wide range of ICRL methodologies, making it a valuable resource for researchers new to the field.
- The clear categorization of approaches based on environments and constraints helps build a structured understanding of ICRL applications.
- I appreciated the discussion on open questions, which sheds light on both short-term and long-term research directions for the community.

**Weaknesses**

- While the survey categorizes the methodologies effectively, it could benefit from a more analytical comparison of these approaches, highlighting their relative strengths and weaknesses.
- Some of the challenges and limitations mentioned are well-known in the broader reinforcement learning literature and aren't unique to ICRL. Including relevant references to existing discussions or explaining why current solutions aren't applicable would strengthen the paper.
- A minor point: some terminology is used inconsistently across sections—for example, "permissibility functions"—which might confuse readers.

---

> ### Author Response · Authors · 2024-11-27
> **Author Response to Reviewer PDYr - Part 1**
>
> Dear Reviewer, we would like to express our sincere gratitude to you for taking the time and effort to review our paper. Your valuable insights and constructive feedback have been instrumental in enhancing the quality and clarity of our work. We hope the following response can resolve your concerns.
>
> > *Comment 1. While the survey categorizes the methodologies effectively, it could benefit from a more analytical comparison of these approaches, highlighting their relative strengths and weaknesses.*
>
> **Response 1.** Thank you for providing this insightful suggestion.
>
> The strengths and weaknesses of the ICRL methods discussed are closely tied to the specific settings and environments they target. To facilitate a more thorough analytical comparison, **we have added a motivational discussion at the beginning of each section** that outlines the limitations of previous methods and the necessity of considering the extension. Please check the details in Sections 4 to 8 (the modifications are **marked in orange color**).
>
> Additionally, we have **expanded the discussion of benchmarks** by including more environments with different kinds of constraints according to some recent studies.
>
> Besides, we have **included a high-level summary of the empirical performance of the ICRL methods** across various benchmarks. Please check the details in **Section 9** (the modifications are marked in **orange color**).
>
> However, extending the discussion to include numerical results presents significant challenges. The ICRL methods are primarily based on diverse benchmarks and settings, which complicates the task of comparing their empirical performance analytically within a single article.
>
> ---
> > *Comment 2. Some of the challenges and limitations mentioned are well-known in the broader reinforcement learning literature and aren't unique to ICRL. Including relevant references to existing discussions or explaining why current solutions aren't applicable would strengthen the paper.*
>
> **Response 2.** Thanks for providing us with the suggestion. Since ICRL is a specialized area within the broader field of the reinforcement learning community, it shares some challenges similar to classical RL. However, ICRL integrates both constrained inference and inverse optimization into the RL framework, which introduces significantly greater complexities compared to traditional RL-based solvers. We have revised our paper by **including detailed evidence about the significance of the challenges and relevant works** to support our argument. Please check the details in **Section 10** (the modifications are marked in **orange color**).
>
> ---
> > *Comment 3. A minor point: some terminology is used inconsistently across sections—for example, "permissibility functions"—which might confuse readers.*
>
> **Response 3.** We apologize for any inconvenience caused. The term "permissibility functions" was previously used to assess the feasibility of agents' movements under specific states or throughout entire trajectories. For instance, it defined the probability that executing an action $a$ under a state $s$, represented as $\phi(s,a)$ and $\mathbb{1}^{\mathcal{M}_{c}}(s,a)$. We have revised this term by **renaming it as a feasibility function**.
>
> ---
> > *Comment 4. At the end of Section 2.4, could you clarify whether "Interpretable Constraints" refers to a regularization method? Is $\phi(s, a)$ a learned model or a fixed function? How does learning $\phi(s, a)$ differ from learning $c(s, a)$, and how does this relate to interpretability?*
>
> **Response 4.** "Interpretable Constraints" is not a regularization method directly in the traditional sense. Instead, it highlights the importance of **designing constraints that are inherently interpretable**, for example, 1) allowing the cost function to have physical significance or 2) understanding which feature is most influential to the changes of cost functions.
>
> Regarding $\phi(s,a)$, it is indeed a learnable model that can be interpreted as a feasibility function. This flexibility allows $\phi(s,a)$ to guide the design and interpretation of cost functions in multiple ways:
>
> - **Probabilistic cost interpretation.** If we define $c(s,a)=1-\phi(s,a)$, the cost function $c(s,a)$ is linear and bounded in $[0,1]$. In this interpretation, the cost represents the probability of being unsafe. This approach is particularly intuitive, as it directly ties the feasibility score to the likelihood of safety violations.
>
> - **Logarithmic cost interpretation.** Alternatively, we can define $c(s,a)=-\log\phi(s,a)$. Under this definition, the cost function $c(s,a)$ is mapped to $[0,\infty]$. Here, the cost is interpreted as a penalty that increases logarithmically as the feasibility score $\phi(s,a)$ approaches zero. This representation is well-suited for cases where infeasibility grows exponentially more severe as safety diminishes.
>
> Therefore, the choice between these definitions is closely tied to the interpretability of the cost function.

---

> ### Author Response · Authors · 2024-11-27
> **Author Response to Reviewer PDYr - Part 2**
>
> > *Comment 5. In Section 3.2, you mention that "ICRL can incorporate recent advancements in RL to learn model-free controllers without knowing or estimating the model dynamics (i.e., model-free methods)." Does this imply that ICRL focuses only on model-free methods, or does it also consider model-based methods?*
>
> **Response. 5.** Thank you for your question. In essence, ICRL is **not limited to model-free approaches**; it has the flexibility to integrate both model-based and model-free methods. In the mentioned sentence, our intent was to highlight its capability to operate in a model-free setting. We have revised the paper to provide a clearer description in **Section 3.2**, which is **highlighted in orange**.
>
> ---
> > *Comment 6. Please clarify the terminology in Section 4.2. Is $\phi_\omega(s,a)$ a permissibility function? Earlier, a permissibility function is defined as "a function learned to distinguish between feasible and infeasible states within these regions." Does the concept of a permissibility function apply only to states, or can it also apply to state-action pairs?*
>
> **Response 6.** In Section 4.2, $\phi_\omega(s,a)$ is indeed a permissibility (or feasibility) function. While the earlier description (in Section 3.2) describes a permissibility function as distinguishing between feasible and infeasible states in their works, **the concept is not limited to states alone**. It can also be extended to state-action pairs, depending on the requirements of the application or environment. We have revised the paper to explicitly state that a permissibility function **can be defined either at the state level or at the state-action pair level in Section 4.2 in orange.**
>
> ---
> > *Comment 7. In Section 6, the issue of epistemic uncertainty seems closely related to the false correlation problem in offline RL. It would be beneficial to add a brief discussion and reference to the false correlation problem.*
>
> **Response 7.** Thank you for this valuable comment. We have **revised Section 6 to include a brief explanation of the false correlation problem** in offline RL and its relevance to epistemic uncertainty in ICRL, which is **marked in orange**.
>
> ---
> > *Comment 8. In Section 8, you mention three levels of Multi-Agent ICRL. Could you elaborate on the key differences between the third level and the first two? Do the first two levels consider agents operating sequentially rather than simultaneously?*
>
> **Response 8.** Thank you for raising this topic. The multi-level framework of Multi-Agent ICRL is designed to address varying complexities and assumptions about agent interactions and roles.
>
> In the first level, Multi-Agent ICRL infers a shared constraint for homogeneous agents. This approach assumes that all agents are similar and face the same constraints, simplifying the inference process.
>
> In contrast, the second level of ICRL is tailored for heterogeneous agents, inferring different constraints for each agent. This level acknowledges and accommodates the diversity in agent capabilities and objectives, allowing for an individualized understanding of constraints.
>
> The key differences between the third level and the first two primarily lie in the **assumptions regarding the forward control policy**. The first two levels treat all other agents as part of the background, essentially assuming that the policy is designed for one agent in isolation, without considering interactions among agents. In contrast, the third level explicitly models the interactions between agents, whether cooperative or competitive, to generate trajectories. This approach aims to optimize the joint behavior of multiple agents, considering their relationships and dynamics, which provides a more comprehensive and realistic simulation of multi-agent environments.
>
> ---
> > *Comment 9. In Section 10, you discuss open questions in ICRL. It would be helpful to cite some papers that have attempted to address these questions, even if they aren't specific to ICRL, since these challenges are common in other RL domains.*
>
> **Response 9.** Thank you for your valuable suggestions. We have revised the paper to **include several references** related to the problem within RL domains in **Section 10, highlighted in orange**.
>
> ---
> We sincerely hope that the response above addresses your concerns. We are deeply grateful for your invaluable feedback and the thoughtful effort you have devoted to reviewing our paper. Thank you once again!

---

> ### Author Response · Authors · 2024-12-10
> **Kind reminder regarding your feedback on TMLR paper 3369**
>
> Dear Reviewer PDYr,
>
> We hope this message finds you well. We understand this might be a particularly busy time for you, and we genuinely appreciate the time you dedicate to reviewing.
>
> We would like to kindly remind you to review our rebuttal at your convenience. Your valuable feedback is instrumental in helping us refine and improve our paper. Should there be any specific points that require further clarification, please do not hesitate to let us know.
>
> Thank you for your time, effort, and thoughtful insights. We sincerely appreciate your contributions to this process.
>
> Best regards,
>
> Authors of TMLR Paper 3369

---

### Review · Reviewer_nG5E · 2024-11-18

**Summary Of Contributions:**

The paper surveys the latest advances in ICRL, providing definitions, key challenges, and applications in areas like autonomous driving, robot control, and sports analytics, aiming to bridge theoretical understanding with practical applications.

**Audience:**

Yes

**Claims And Evidence:**

Yes

**Requested Changes:**

I request the authors can restructure the paper to improve clarity and cohesion, add a mathmatical problem fomulation of ICRL.

**Strengths And Weaknesses:**

My expertise does not include inverse constrained reinforcement learning (ICRL), so I approached this manuscript with the intent to learn more about the topic. However, I found the paper challenging to follow for a reader without a background in ICRL. Firstly, it seems that this paper is based on a finite-horizon MDP, which the authors should make clear. There are also other settings, such as infinite-horizon MDPs and POMDPs, which should ideally be discussed in a review. This led me to wonder: is ICRL only applicable to finite-horizon MDPs?

Furthermore, I noticed that the optimization problem in equation (1) includes a policy entropy term, which is not that common in general reinforcement learning problems, and no reference was provided by the authors. Has this problem formulation become a standard paradigm in ICRL? Although the authors discuss the benefits later, they do not directly address this question. For these two fundamental points, I was unable to find answers in the manuscript. This indicates that the survey paper does not extend the discussion but simply summarizes the current work.

The survey paper's structure also makes it difficult for readers to follow. The transitions between sections are weak, and the connections between different parts are not well-established, making the overall flow feel disjointed. More improvment to writing quality and structural coherence is needed. For instance, while Figure 3 summarizes the main procedure of ICRL, I could not find an introduction to the mathematical model of ICRL. What is the core optimization problem that ICRL aims to solve?

Starting from Section 4, titled "Constraint Inference in the Deterministic Environment," the writing becomes increasingly challenging for readers, with extensive explanations lacking sufficient elaboration. The taxonomy of different categories is also not clearly explained or compared. I hope the authors can restructure the paper to improve clarity and cohesion.

The paper does have its strengths. It introduces an emerging topic that is gradually gaining attention in the academic community, which makes it a timely and relevant contribution.

---

> ### Author Response · Authors · 2024-11-27
> **Author Response to Reviewer nG5E - Part 1**
>
> Dear Reviewer, we apologize for any confusion caused by our paper. As ICRL is an emerging research topic, there currently lacks a structured survey of the relevant literature. We have dedicated considerable time and effort to summarize recent advancements in a structured and principled manner. We appreciate your concerns and would like to provide the following clarifications.
>
> > *Comment 1. Firstly, it seems that this paper is based on a finite-horizon MDP, which the authors should make clear. There are also other settings, such as infinite-horizon MDPs and POMDPs, which should ideally be discussed in a review. This led me to wonder: is ICRL only applicable to finite-horizon MDPs?*
>
> **Response 1.** Similar to the commonly studied RL settings, ICRL is based on **Episodic MDPs, where the MDP terminates at a time step $T$, though this planning horizon $T$ is not fixed**. For example, the game may 1) terminate when the agent reaches a terminating or goal state, or 2) assign a terminating probability associated with each state. Under the Episodic MDP, the planning horizon runs from 0 to infinity, e.g., $t\in[0,\infty)$. **We have introduced this in Section 2.1, which is highlighted in red.**
>
> The assumption of **Episodic** MDPs offers several key advantages:
>
> - It eliminates the dependence of policy and value functions on the planning time step $ t $, significantly simplifying the modeling process. In contrast, assuming finite-horizon MDPs would require both the policy and value functions to be dependent on $t$. For example, under finite-horizon MDPs, the size of action-value functions' $Q_t(s,a)$ space is $|\mathcal{S}|\times|\mathcal{A}|\times\mathcal{T}$.
>
> - It facilitates a straightforward definition of the probability of generating a trajectory, expressed as $p(\tau) = \prod_{t=0}^{T} \pi(a_t|s_t) p_\mathcal{T}(s_{t+1}|s_t,a_t) $ where $ p(\tau) \in [0,1] $. Conversely, in infinite-horizon MDPs, as $T$ approaches infinity, $ \lim _{T\rightarrow\infty} p(\tau)$ is always zero, which makes the definition of $p(\tau)$ no longer sound.
>
> In the revised version of this paper, **we have clarified the definition of episodic MDPs and fixed the relevant typos.**
>
> ---
> > *Comment 2. Furthermore, I noticed that the optimization problem in equation (1) includes a policy entropy term, which is not that common in general reinforcement learning problems, and no reference was provided by the authors. Has this problem formulation become a standard paradigm in ICRL?*
>
> **Response 2.** We apologize for any confusion caused.
>
> The concept of **maximum entropy has been extensively explored in the field of Inverse Reinforcement Learning (IRL) research** [1], as it provides a probabilistic representation of the optimal policy. This approach facilitates a maximum likelihood method for updating reward functions. Inverse Constraint Reinforcement Learning (ICRL) [2] adopts this paradigm from IRL by incorporating an entropy term into the objective for constraint inference.
>
> We include an entropy term in the RL objective to ensure consistency with both IRL and ICRL settings. This inclusion is crucial as the forward process in IRL and ICRL involves solving a maximum entropy RL objective, as outlined in Equation (1). Besides, the maximum entropy RL objective aligns with recent advancements in the field of RL, such as Soft Q-learning [3] and Soft Actor-Critic [4], which are standard RL solvers widely studied in recent years.
>
> We have **revised our survey to ensure readers can comprehend this idea.**
>
> ---
> > *Comment 3. The transitions between sections are weak, and the connections between different parts are not well-established, making the overall flow feel disjointed. More improvment to writing quality and structural coherence is needed.*
>
> **Response 3.** Thanks for raising this important concern. In response to the concerns raised and as suggested by Reviewer PDYr, we have **incorporated a motivational discussion at the beginning of each section.** This discussion outlines the limitations of previous methods and explains the necessity of considering extensions to these methods. This approach ensures that the reader understands the context and rationale behind our research contributions (check the words **marked in orange color**).
>
> In addition, we **added a diagram to our paper (See Figure 6)**, summarizing the sections of our survey and their connection in a structured manner. We hope it can improve the clarity of our survey.
>
> ---
> **References**
>
> [1] Ziebart, Brian D., et al. "Maximum entropy inverse reinforcement learning." AAAI, 2008.
>
> [2] Malik, Shehryar, et al. "Inverse constrained reinforcement learning." ICML, 2021.
>
> [3] Haarnoja, Tuomas, et al. "Reinforcement learning with deep energy-based policies." ICML, 2017.
>
> [4] Haarnoja, Tuomas, et al. "Soft actor-critic algorithms and applications." arXiv, 2018.

---

> ### Author Response · Authors · 2024-11-27
> **Author Response to Reviewer nG5E - Part 2**
>
> > *Comment 4. For instance, while Figure 3 summarizes the main procedure of ICRL, I could not find an introduction to the mathematical model of ICRL. What is the core optimization problem that ICRL aims to solve?*
>
> **Response 4.** Thank you for raising this point. Although we have introduced the **main processes of ICRL in Section 2.3 and Figure 3**, we agree that adding a mathematical model could better formulate the problem. Essentially, ICRL can be modeled as a tri-level optimization problem [5]. Specifically, previous ICRL solvers explicitly model constraints and infer the cost function by alternatively optimizing the policy and the constraint function.
> In the phase of policy optimization, they commonly solve a CRL problem by studying its Lagrangian dual:
>
> $\arg\min_{\pi} -\mathbb{E}_{{\rho}_M^\pi(s, a)}\left[ r(s, a) - \lambda^* c(s, a) - \frac{1}{\alpha} \log \pi(a|s) \right] - \lambda^* \epsilon$
>
> where $\lambda^*$ denotes the optimal Lagrange multiplier. Based on the dual representation of CRL problem, ICRL solvers involve learning the cost functions $c$ from an expert demonstration dataset $\mathcal{D}_{E}$. This is essentially solving a tri-level optimization problem [5]:
>
> $\max_c \max_\lambda \min_\pi \mathbb{E}_{(s_E, a_E) \sim \mathcal{D}_E}[ r(s_E, a_E) - \lambda c(s_E, a_E)- \frac{1}{\alpha} \log \pi(a_E | s_E)] -$
>
> $\mathbb{E}_{(s, a) \sim \rho^\pi_M(s, a)} [ r(s, a) - \lambda c(s, a) - \frac{1}{\alpha} \log \pi(a | s) ] - \lambda \epsilon$
>
> We have revised the paper to make it clearer.
>
>
> [5] Kim, Konwoo, et al. "Learning shared safety constraints from multi-task demonstrations." NeurIPS, 2024.
>
> ---
> > *Comment 5. Starting from Section 4, titled "Constraint Inference in the Deterministic Environment," the writing becomes increasingly challenging for readers, with extensive explanations lacking sufficient elaboration. The taxonomy of different categories is also not clearly explained or compared. I hope the authors can restructure the paper to improve clarity and cohesion.*
>
> **Response 5.** We apologize for any confusion caused.
>
> To clarify the relationships between each section starting from Section 4, we have **added a diagram (Figure 6)**. In summary, we begin with the simplest ICRL setting in Section 4.1, where we assume deterministic dynamics and a discrete state-action space. Over time, we gradually generalize to more realistic settings, beginning with stochastic dynamics (Section 5), then addressing demonstration data of limited size (Section 6), unknown reward functions (Section 7), and handling scenarios involving multiple agents (Section 8).
>
> In this structured and principled approach, **we include most aspects of ICRL research**. This allows readers to start with the most fundamental setting of ICRL, which is based on maximum entropy principles, and observe how the framework is generalized to accommodate more complex settings required by various applications. More importantly, readers can easily navigate our survey and select the sections that align most closely with their research interests by referencing the topics covered in each section. This approach enhances the utility of our survey, making it a valuable resource for researchers at different stages of their ICRL exploration.
>
> **We have taken your comments into account and revised the paper accordingly**, adding detailed explanations in each section to enhance the overall coherence and depth of our discussion on ICRL. **We also received some encouraging comments from other reviewers**, for example, Reviewer 4Ecj: "The paper is well structured" and Reviewer PDYr: "The clear categorization of approaches based on environments and constraints helps build a structured understanding of ICRL applications." **We believe the survey has become more clear and readable after incorporating your valuable suggestions.**
>
> ---
> We deeply appreciate your insightful concerns and the time you have taken to review our work. Your feedback has provided us with an invaluable opportunity to refine and clarify our paper. Thank you once again for your thoughtful comments and for helping us improve the quality and clarity of our work.

---

> ### Author Response · Authors · 2024-12-10
> **Kind reminder regarding your feedback on TMLR paper 3369**
>
> Dear Reviewer nG5E,
>
> We hope this message finds you well. We understand this might be a particularly busy time for you, and we genuinely appreciate the time you dedicate to reviewing.
>
> We would like to kindly remind you to review our rebuttal at your convenience. Your valuable feedback is instrumental in helping us refine and improve our paper. Should there be any specific points that require further clarification, please do not hesitate to let us know.
>
> Thank you for your time, effort, and thoughtful insights. We sincerely appreciate your contributions to this process.
>
> Best regards,
>
> Authors of TMLR Paper 3369

---

### Author Response · Authors · 2024-11-29
**Revised Manuscript Submission and Further Discussion**

Dear Reviewers,

Thank you for your valuable feedback on our article. In the revised manuscript, for ease of review, we have highlighted the changes made in response to each reviewer's comments: 1) reviewer nG5E in red, 2) reviewer PDYr in orange, and 3) reviewer 4Ecj in blue. We believe that the revisions and our response address the main concerns you raised. Your expertise has been instrumental in enhancing the quality of our work.

Should you have any further comments or require additional discussion, we are eager to engage with you to refine the manuscript further.

Thank you again for your time and thoughtful consideration.

Best regards,

Authors of TMLR Paper 3369

---

### Decision · Action_Editor_qm5u · 2024-12-20

**Recommendation:** Accept as is

**Comment:**

While the reviewer nG5E still has some concerns (around the problem formulation), I believe that the benefits outweigh the downside of publishing the paper.

Reviewer 4Ecj: "As far as I know this is the first survey paper in the area of constraint inference in RL. I think that the authors presented most of the up to date work in this area. Although I personally prefer surveys that also review the approaches quantitatively, I understand that in this case this might be trickier given that there is high variability in the setups for which each method was developed."

Reviewer PDYr: "I recommend accepting this submission. The authors have made valuable improvements in clarifying key concepts and addressing the reviewer's requests, which improves the paper’s readability and applicability. These updates make the work more accessible to a broader audience, thereby increasing its potential impact as a survey paper."

Reviewer nG5E: "I read the revised paper again. However, I still have some concerns regarding the problem formulation in the ICRL framework, particularly the inclusion of entropy in the objective function. While the manuscript emphasizes the benefits of incorporating entropy, it lacks a fundamental and substantive comparison with approaches that do not include entropy. This comparison is crucial to justify the rationale for using entropy at a deeper, foundational level."

**Audience:**

The paper addresses an important intersection of the safety, and reinforcement learning being relevant to both research communities, as well as the application areas in the domains of the safety critical control systems.

**Claims And Evidence:**

The paper presents a survey on Inverse Constrained Reinforcement Learning. The survey includes the problem's definition, and the methods and challenges are categorized based on environment type.

The reviewers find the survey timely, and contemporarily relevant. While a quantitative comparison of the methods would have been highly desirable, the authors addressed the impracticality of such comparison. One reviewer still has reservations around the inclusion of entropy in the problem formulation.

Nevertheless, the survey as it stands would bring valuable resource to the emerging field, and publishing outweighs the downside of not publishing. The follow up work from the community can address quantitive comparisons and bring alternative problem formulations -- and this work can help push the field forward.